# Interethnic analyses of blood pressure loci in populations of East Asian and European descent

Fumihiko Takeuchi, Masato Akiyama, Nana Matoba, Tomohiro Katsuya, Masahiro Nakatochi, Yasuharu Tabara et al.[#]

Blood pressure (BP) is a major risk factor for cardiovascular disease and more than 200 genetic loci associated with BP are known. Here, we perform a multi-stage genome-wide association study for BP (max $N = 289{,}038$) principally in East Asians and meta-analysis in East Asians and Europeans. We report 19 new genetic loci and ancestry-specific BP variants, conforming to a common ancestry-specific variant association model. At 10 unique loci, distinct non-rare ancestry-specific variants colocalize within the same linkage disequilibrium block despite the significantly discordant effects for the proxy shared variants between the ethnic groups. The genome-wide transethnic correlation of causal-variant effect-sizes is 0.898 and 0.851 for systolic and diastolic BP, respectively. Some of the ancestry-specific association signals are also influenced by a selective sweep. Our results provide new evidence for the role of common ancestry-specific variants and natural selection in ethnic differences in complex traits such as BP.

---

High blood pressure is a major risk factor for cardiovascular disorders such as coronary heart disease and stroke. Approximately 10 million deaths each year can be attributed to high blood pressure globally[1,2]. An individual's risk for high blood pressure is determined by genetic, environmental and demographic factors and their interaction. Genome-wide association studies (GWASs) and/or large-scale analyses by gene-centric (or exome) variation arrays have identified over 200 genetic loci influencing blood pressure in predominantly European-descent populations (henceforth referred to as Europeans)[3–8]. The prevalence of high blood pressure is increased in people of East Asian ancestry, contributing to their increased risk of stroke[9]. The reasons for such ethnic differences remain to be clarified from the viewpoint of genetic susceptibility as well as lifestyle. Although the recent progression of GWAS in East Asians allows us to make a preliminary comparison of association signals between the populations[10,11], the sample sizes of GWAS in East Asians have been generally much smaller than those in Europeans and under-powered for the comprehensive interethnic comparison at a genome-wide scale. Therefore, large-scale genome-wide association data in both ethnic groups are required for systematic, genome-wide interethnic comparison.

Here, we perform a multi-stage GWAS with a discovery sample of 130,777 East Asian individuals and follow-up meta-analyses involving East Asians and Europeans (max $N = 289,038$), to seek both transethnic and ancestry-specific genetic effects for five blood pressure phenotypes: systolic blood pressure (SBP), diastolic blood pressure (DBP), pulse pressure (PP), mean arterial pressure (MAP), and hypertension. We then seek interethnic genetic heterogeneity of GWAS results between East Asians and Europeans, followed by examination of natural selection as a potential mechanism underlying the ethnic differences in genetic susceptibility for blood pressure as well as other complex traits. We report ancestry-specific blood pressure variants and selection signals in this study.

## Results

**Genome-wide association analysis and lookup for replication.** Adopting a joint analysis strategy[12], we performed a GWAS, which consisted of stage 1 (discovery) and stage 2 (follow-up), and a replication study (Supplementary Fig. 1). In stage 1 of GWAS, we used genome-wide association data from 130,777 individuals of Japanese ancestry. Characteristics of participants, genotyping arrays, and imputation are summarized in Supplementary Tables 1, 2. Genomic control and intercepts from linkage disequilibrium (LD) score regression[13] were calculated at each study level ($\lambda_{GC} = 0.89–1.24$ and LD Score regression intercept = $0.94–1.06$), indicating no residual confounding biases such as population stratification (Supplementary Table 2). Since the LD Score regression intercept can account for polygenic effects and inflation due to large sample size[13], we applied the LD Score regression intercept as a correction factor for cohorts with a sample size of >3000 individuals (BBJ in this study). Genomic control $\lambda_{GC}$ was used as a correction factor in the other studies. Quantile–quantile plots for each of the five blood pressure traits are presented in Supplementary Fig. 2. Phenotype-specific meta-analysis was carried out in the two-stage approach for both the East Asian-specific and transethnic meta-analyses (Supplementary Figs. 1, 3). Genome-wide association results in the stage-1 identified 13,003 SNPs with a P value < $1.6 \times 10^{-5}$ against any blood pressure phenotype in East Asians. This set of 13,003 SNPs (sentinel SNPs listed in Supplementary Data 1) was followed up in 53,008 East Asian individuals (stage 2). Additionally, these 13,003 SNPs were examined in the transethnic stage with phenotype-specific results for Europeans (max $N = 105,253$) from

the International Consortium on Blood Pressure (ICBP) GWAS ($N = 69,909$)[3] and the International Genomics of Blood Pressure (iGEN-BP) Consortium ($N = 35,344$)[10]; there was no overlap in samples between the two data sets. Sentinel SNPs (smallest P value against any blood pressure phenotype) that (i) reached $P < 5 \times 10^{-8}$ in combined meta-analysis of stages 1 and 2 and (ii) showed evidence of support ($P < 0.05$) in the stage 2 meta-analysis alone are reported as novel loci in this study. We identified 19 previously unreported loci; 15 loci in East Asian-specific analyses and 4 additional loci in the transethnic meta-analysis (Table 1 and Supplementary Data 2). By lookup in an independent replication sample of Europeans from the UK Biobank ($N = 422,771$)[14] plus East Asians from the China Kadoorie Biobank ($N = 94,201$)[15], we examined associations at our list of 19 sentinel SNPs. With the exception of four SNPs, 15 sentinel SNPs showed significant ($P < 0.00263 = 0.05/19$) blood pressure association with the concordant direction of allelic effects (Supplementary Data 2), thus validating the loci.

Regional association plots are shown for the 19 newly identified loci in Supplementary Fig. 4. Associations of the 19 sentinel SNPs with other blood pressure phenotypes are demonstrated in Supplementary Data 3. In the discovery stage, we also replicated blood pressure associations at previously reported loci, which included 36 loci at genome-wide significance and further 179 loci at nominal significance ($P < 0.05$) (Supplementary Data 4).

**Functional annotations for new loci.** To identify candidate genes at the newly identified blood pressure loci, we examined whether any of the association signals (sentinel blood pressure SNP and SNPs in East Asian LD $r^2 > 0.80$) were coding or associated with gene expression and other traits. At three loci, the sentinel SNPs were nonsynonymous, and 4 of 19 novel loci contained SNPs (in LD of $r^2 > 0.80$ with the top eVariant) associated with expression quantitative trait loci (eQTLs) in at least one tissue in the Genotype-Tissue Expression (GTEx) database (Supplementary Tables 3–5). At two candidate gene loci, proxy SNPs (rs760077 at *MTX1* and rs3825942 at *LOXL1*) were nonsynonymous and associated with eQTLs. Furthermore, seven sentinel SNPs and/or their proxy SNPs ($r^2 \geq 0.95$) were previously reported to be significantly associated with non-blood pressure traits (Supplementary Data 5), including a sentinel SNP (rs11642015) at the *FTO* locus on 16q22, whose proxies ($r^2 = 0.97–0.99$) have been reported to associate with body mass index and type 2 diabetes[16]. In our study, rs11642015 was significantly associated with SBP, MAP, and PP ($P = 1.9 \times 10^{-12} – 1.3 \times 10^{-9}$) with consistent reproducibility in both stages of East Asian analyses (Supplementary Data 1, 3). In addition, rs11642015 was recently identified to be significantly associated with SBP in multi-ancestry GWAS meta-analysis incorporating gene−smoking interaction[17].

**Interethnic heterogeneity of GWAS results.** In the present study, the availability of genome-wide association data from >100,000 individuals for both East Asians and Europeans separately motivated us to perform additional analyses of systematic, genome-wide interethnic comparison. We used transethnic association summary statistics available for both East Asian ($N = 158,645$ from stage 1 and iGEN-BP) and European (max $N = 105,253$ from ICBP and iGEN-BP) GWAS results in the subsequent analysis of interethnic heterogeneity. We defined interethnic heterogeneity as heterogeneity of genetic (or allelic) impact on SBP between the ethnic groups. Using GWAS data sets, we compared the genetic impact at transethnic SNPs and detected a total of eight interethnic heterogeneity loci—two significant ($P < 5 \times 10^{-8}$) and six suggestive ($5 \times 10^{-8} \leq P < 1 \times 10^{-6}$) loci (Fig. 1a and Supplementary Data 6). In this study we distinguished the allelic

**Table 1 Genetic loci newly identified to be associated with blood pressure**

| Sentinel SNP | Chr | Position | EA/NEA | EAF | Trait | N | Effect | P |
|---|---|---|---|---|---|---|---|---|
| **Genome-wide significant and replicated** | | | | | | | | |
| rs2990220 | 1 | 155,190,254 | A/T | 0.83 | MAP | 183,654[a] | −0.41 (0.06) | $2.2\times10^{-12}$ |
| rs6772151 | 3 | 46,896,499 | A/C | 0.29 | DBP | 156,503[a] | 0.28 (0.05) | $7.8\times10^{-9}$ |
| rs17622152 | 3 | 183,520,112 | A/G | 0.47 | MAP | 183,759[a] | −0.25 (0.04) | $2.0\times10^{-8}$ |
| rs12209106 | 6 | 1,621,042 | T/G | 0.68 | DBP | 160,436[a] | 0.28 (0.05) | $6.4\times10^{-9}$ |
| rs78399431 | 7 | 1,141,470 | A/G | 0.24 | MAP | 179,411[a] | 0.30 (0.05) | $9.6\times10^{-9}$ |
| rs2125067 | 10 | 48,434,420 | C/G | 0.12 | SBP | 179,003[a] | 0.60 (0.10) | $4.8\times10^{-9}$ |
| rs2305013 | 11 | 120,340,060 | A/T | 0.85 | SBP | 180,894[a] | −0.59 (0.09) | $5.6\times10^{-10}$ |
| rs5006548 | 12 | 32,692,233 | T/G | 0.16 | HT | 71,847[a] | 0.09 (0.02) | $2.2\times10^{-8}$ |
| rs1535464 | 14 | 100,793,431 | A/G | 0.10 | SBP | 183,690[a] | −0.61 (0.10) | $3.5\times10^{-9}$ |
| rs66978877 | 19 | 18,455,657 | T/C | 0.55 | HT | 68,850[a] | 0.07 (0.01) | $4.5\times10^{-9}$ |
| rs6021247 | 20 | 50,108,980 | A/G | 0.58 | SBP | 183,785[a] | 0.37 (0.06) | $5.0\times10^{-9}$ |
| rs3853476 | 5 | 141,817,754 | A/G | 0.58 | MAP | 244,831[b] | −0.20 (0.03) | $6.0\times10^{-9}$ |
| rs10821808 | 10 | 62,390,646 | A/G | 0.58 | SBP | 288,917[b] | −0.29 (0.05) | $3.4\times10^{-9}$ |
| rs4418728 | 10 | 94,839,724 | T/G | 0.62 | DBP | 256,118[b] | −0.20 (0.03) | $1.5\times10^{-8}$ |
| rs1078967 | 15 | 74,222,987 | T/C | 0.15 | SBP | 265,280[b] | 0.42 (0.07) | $5.6\times10^{-9}$ |
| **Genome-wide significant but not replicated** | | | | | | | | |
| rs2076460 | 1 | 27,972,058 | C/G | 0.30 | SBP | 174,846[a] | −0.42 (0.07) | $3.6\times10^{-9}$ |
| rs11642015 | 16 | 53,802,494 | T/C | 0.21 | SBP | 174,917[a] | 0.58 (0.08) | $1.9\times10^{-12}$ |
| rs9303509 | 17 | 64,530,887 | A/C | 0.40 | SBP | 183,769[a] | 0.37 (0.06) | $3.9\times10^{-9}$ |
| rs66658258 | 20 | 61,462,502 | C/G | 0.58 | DBP | 164,638[a] | 0.28 (0.05) | $1.0\times10^{-8}$ |

Position is Build 37; EA: effect allele; NEA: non-effect allele; EAF: effect allele frequency; N: sample size ([a]East Asians only; [b]with European follow-up samples); Effect: as unit change in blood pressure (SE) per effect allele copy (SBP, DBP, PP, MAP) or as log odds ratio per effect allele (HT)

impact from allelic effect-sizes as previously defined by Brown et al.[18]; allelic impact is the genotype−phenotype correlation coefficient, which is approximately a product of allelic effect and minor allele frequency (MAF). Seven of the eight loci with interethnic heterogeneity were annotated to the previously reported blood pressure loci; sentinel blood pressure SNPs at half of them (i.e., four loci near the *CACNB2*, *C10orf107*, *SH2B3* and *DPEP1* genes[3,5]) were found to be in LD ($r^2 \geq 0.2$) with the SNPs showing some evidence for interethnic heterogeneity. The two loci with significant interethnic heterogeneity were on 12q24 and 10q21 and both contained multiple association signals (Fig. 1b). For the region on 12q24 spanning 1.5 Mb, two independent association signals, each specific to Europeans (near rs3184504 at *SH2B3*) and East Asians (near rs671 at *ALDH2*), had been identified[19]. We found that both of the signals were responsible for the discordant direction of allelic effects on 12q24 (Fig. 1c and Supplementary Fig. 5a, b). Similarly, we observed two independent association signals near the *C10orf107* transcript on 10q21.2 (Fig. 1c and Supplementary Fig. 5c, d). The derived alleles of ancestry-specific sentinel SNPs on 10q21 (rs4590817 and rs145193831 specific to Europeans[3] and East Asians respectively) arose from a haplotype shared between ethnic groups, containing multiple transethnic SNPs. The discordant direction of effects for the shared haplotypes could be explained by alternation of effects attributable to the derived alleles of rs4590817 (decreasing in Europeans) and rs145193831 (increasing in East Asians) (Supplementary Fig. 6 and Supplementary Data 7).

**Ancestry-specific SNP loci**. A total of 750 previously reported SNPs (listed in Supplementary Data 4) plus 19 newly identified SNPs could be classified into 485 loci by regarding two SNPs at most 500 kb apart to belong to the same locus. After exclusion of 39 loci (MAF < 0.01 in both East Asians and Europeans, or no data available in GWAS data sets for both populations), 446 loci were retained and categorized into two groups—group 1 and group 2. Group 1 consisted of 382 loci with MAF ≥ 0.01 in both populations and group 2 consisted of 64 loci with potential ethnic

specificity, i.e., MAF < 0.01 in either East Asians or Europeans. Group 2 was further classified into group 2a (46 loci with MAF < 0.01 in one population and MAF ≥ 0.05 in the other) and group 2b (18 loci with MAF < 0.01 in one population and 0.01 ≤ MAF < 0.05 in the other) (Supplementary Fig. 7).

With regards to interethnic heterogeneity of association signals, we assumed two distinct scenarios: whether the underlying causal variants are shared between the ethnic groups or not. However, due to substantial interethnic differences in LD structure, it is not always feasible to distinguish between the two. First, as an example of the potential nonshared causal variant (or ancestry specificity), we examined interethnic comparability of genetic impact on blood pressure at 48 loci (46 loci in group 2a plus 2 target loci with potential ancestry specificity—*C10orf107* and *CACNB2*—included in group 1; Supplementary Fig. 7), where sentinel common (MAF ≥ 0.05) blood pressure SNPs originally reported in a given ethnic group were monomorphic or MAF < 0.01 in the second ethnic group[3–8,19]. Then, we investigated interethnic heterogeneity at non-rare (MAF ≥ 0.01 in both ethnic groups) blood pressure loci (group 1 in Supplementary Fig. 7) that might be shared between the ethnic groups as described later.

Considering the observations on 12q24 and 10q21, we explored common proxy SNPs forming a haplotype shared between ethnic groups at the locus (denoted as haplo-SNPs), for which the most significant interethnic heterogeneity of genetic impact was detected (Supplementary Fig. 8a–c). At a total of 11 loci (or 10 unique loci when the *ALDH2* and *SH2B3* loci on 12q24 were combined) (Supplementary Figs. 6, 9 and Supplementary Data 7), haplo-SNPs showed significant ($P < 1.5\times10^{-4}$ under region-wise correction) heterogeneity between two ethnic groups. At 8 of 11 loci, we found that distinct common ancestry-specific variants colocalized within the same LD block and that the direction of effects for the proxy shared SNPs was discordant between the ethnic groups, similar to 12q24 and 10q21. On 5q14, for instance, a genome-wide significant association of rs112862634 with SBP, DBP, and MAP was detected in East Asians of this study (Supplementary Data 1), while SBP association of rs10059921 was previously reported in its vicinity (456 kb apart from

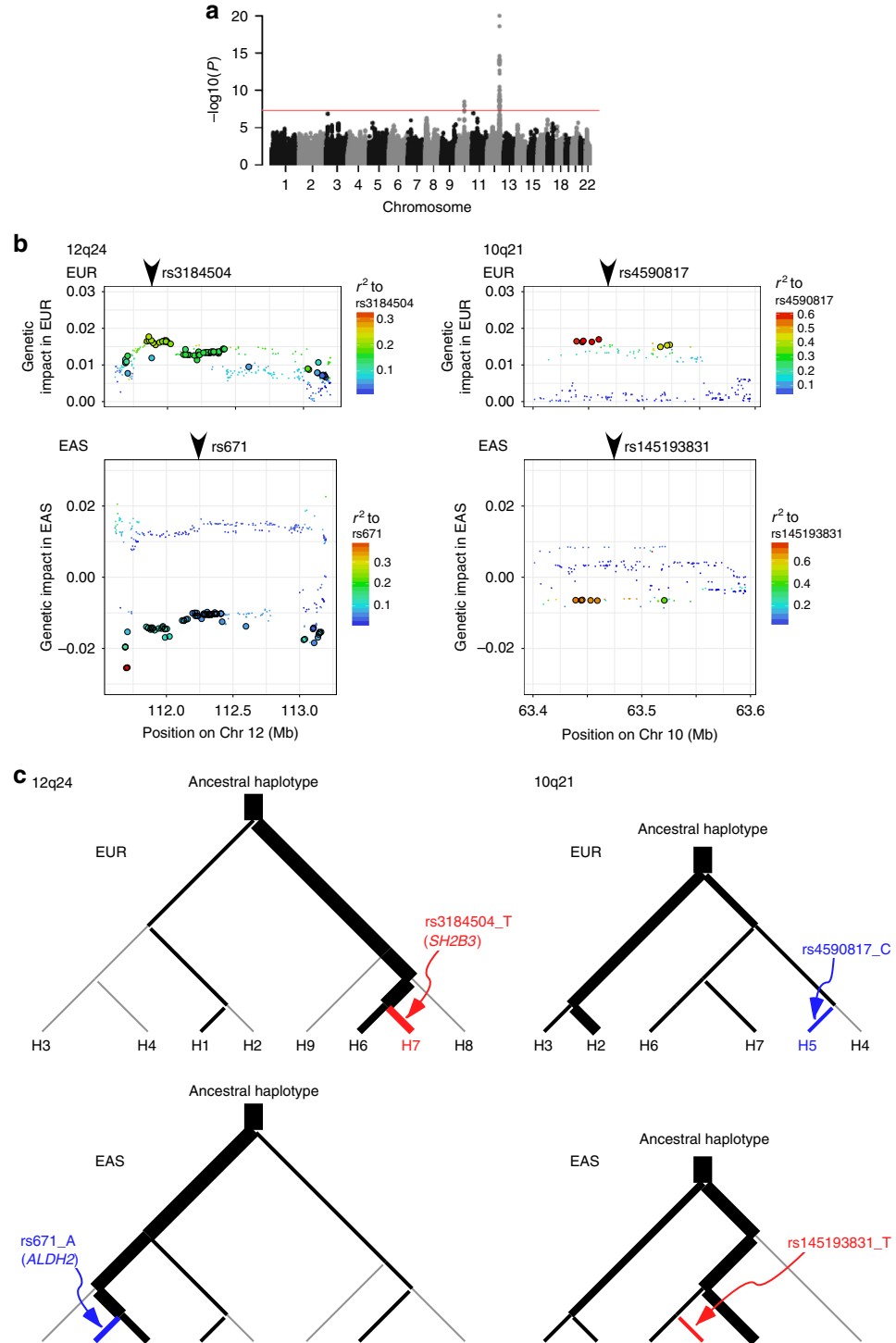

**Fig. 1** Interethnic heterogeneity of genetic impact of SBP. **a** Manhattan plot showing results for genome-wide scan of genetic impact heterogeneity. The genetic impact at transethnic SNPs were compared between two populations of different ancestries using GWAS data sets. **b** Regional plots on 12q24 and 10q21, where there were multiple SNPs with significant ($P < 5 \times 10^{-8}$) evidence for interethnic heterogeneity (see Supplementary Data 6). Bordered circles represent SNPs with significant interethnic heterogeneity. Transethnic SNPs were plotted in two panels at each locus; genetic impacts of each SNP are denoted separately for Europeans (EUR, top panel) and East Asians (EAS, bottom panel) on 12q24 (left) and 10q21 (right) such that genetic impacts in Europeans are positive. In the individual regional plots, the correlation of ancestry-specific sentinel SNP to other SNPs at the locus is shown on a scale from minimal (blue) to maximal (red); the sentinel SNPs thus benchmarked are rs3184504 (EUR specific) and rs671 (EAS specific) on 12q24 and rs4590817 (EUR specific) and rs145193831 (EAS specific) on 10q21. The position of ancestry-specific sentinel SNP is indicated by an arrow head. **c** Phylogenetic relationships of ancestry-specific sentinel SNPs with transethnic haplotypes detectable in Europeans (top) and East Asians (bottom) on 12q24 (left) and 10q21 (right). Each node corresponds to a haplotype and the SNPs appear on the edges. The edge width reflects the haplotype frequency in the corresponding ethnic groups. At each locus, blood pressure increasing and decreasing haplotypes and derived, ancestry-specific alleles are colored in red and blue, respectively

rs112862634) in Europeans[8]. It turned out that rs112862634 was in strong LD (East Asian LD $r^2 = 0.95$) with a haplo-SNP (rs6882046) at this locus and distinct common ancestry-specific variants with mutually inverted genetic effects—European-specific rs10059921 (MAF = 0.09 in EUR) and East Asian-specific rs78245349 (MAF = 0.46 in EAS)—did colocalize in this region (Supplementary Fig. 8a and Supplementary Data 7). At the remaining 3 (of 11) loci, alternate rare ancestry-specific SNPs were likely to exist in the second ethnic group, although they were not detectable in our search of public databases. We designated these as a common ancestry-specific variant association model as discussed below.

We hypothesized that there were three major combinations of East Asian-/European-specific SNPs and their resultant direction of effects for haplo-SNPs forming a shared haplotype at the locus, as schematically shown in Supplementary Fig. 6b. In accordance with this notion, we detected three types in this study (Supplementary Fig. 9), among which the first and major type (32 of 48 loci in group 2a) consisted of the cases with mutually inverted genetic effects as explained above. The second type consisted of those with distinct ancestry-specific variants showing concordant directions of effect such as the *FGR* locus (Supplementary Figs. 8b, 9). The third type consisted of those with distinct ancestry-specific variants showing discordant genetic effects, one of which appeared to be almost neutral such as the *GNAS/EDN3* locus (Supplementary Fig. 8c). However, without using larger sample sizes, it appeared to be difficult to show statistically significant interethnic heterogeneity in particular, for those in the second or third type. At one locus (near *HSD17B1* on 17q21), a haplo-SNP could not be selected (Supplementary Fig. 8d and Supplementary Data 7), presumably because of the ancestry-specific LD structure and the modest strength of association in the index ethnic group (Europeans at the locus) of this study.

For loci with potential ancestry specificity (i.e., MAF < 0.01 in one population and 0.01 ≤ MAF < 0.05 in the other; 18 loci classified as group 2b in Supplementary Fig. 7), we did not investigate interethnic heterogeneity of association signals because of difficulties in the relevant test for rare (MAF < 0.01) and low-frequency (0.01 ≤ MAF < 0.05) genetic variants by using imputed GWAS results[20].

**Heterogeneity at variants polymorphic in both ancestries.** In addition to the ancestry-specific loci, we investigated interethnic heterogeneity at non-rare (MAF ≥ 0.01 in both ethnic groups) blood pressure loci that might be shared between the ethnic groups; 382 tested loci were either previously reported or newly identified in the present study (denoted as group 1 in Supplementary Fig. 7a). Since ICBP and iGEN-BP (European) data were imputed with HapMap SNPs, approximately one-third of group-1 SNPs were unavailable in our European GWAS data sets. Thus, 242 (out of 382) loci in group 1 were subjected to interethnic comparison of genetic impact on a lead blood pressure trait (Supplementary Data 8). Although majority of them appeared to show concordant effects (correlation coefficient $r = 0.754$), nine sentinel SNPs (3.7%) showed significant ($P_{hetero} < 2.1×10^{-4}$) interethnic heterogeneity (Supplementary Fig. 10). Genetic impacts were more prominent in Europeans than in East Asians at eight of nine loci apart from rs1451538 in *SLC28A1*, at which genetic impacts were prominent in East Asians but not in Europeans (Supplementary Data 8). There were no proxy SNPs near each of the eight loci in the same LD block (Supplementary Fig. 11), which could have shown stronger association signals in East Asians than the sentinel SNPs originally reported in Europeans due to potential interethnic differences in LD structure, if

any. Of note is the finding on 10q23 near *PLCE1*, there was another SBP association signal at rs7080472 in East Asians ($P = 3.9×10^{-8}$ in the combined samples; Supplementary Data 1) despite the absence of prominent association at rs932764, whose association was previously reported[4] and prominent in Europeans (Supplementary Fig. 11). rs7080472 was located in the LD block next to the one for rs932764 (East Asian LD $r^2 = 0.003$ between rs7080472 and rs932764). On 10q21 near *C10orf107*, a DBP association signal was previously reported at rs1530440[21], which we found to be in LD (European LD $r^2 = 0.48$) with an ancestry-specific SNP at the locus, rs4590817, aforementioned (Supplementary Data 7). Also, on 10p12 near *CACNB2*, a DBP association signal was previously reported at rs1813353[3], which we found to be in LD (European LD $r^2 = 0.56$) with an ancestry-specific SNP at the locus, rs12258967. These indicated that interethnic heterogeneities identified for non-rare transethnic variants on 10q21 and 10p12 were the cases for which common ancestry-specific variants were actually responsible.

By calibrating the proportion in the group-1 subset, in which blood pressure GWAS results for interethnic comparison were available for 242 (of 382) loci, we estimated the proportion of loci showing significant interethnic heterogeneity within the total blood pressure loci tested (N = 446). The estimated proportion was 2.5% each in group 1 and group 2a, where the *C10orf107* and *CACNB2* loci were counted in group 2a (Supplementary Fig. 7b).

**Genetic correlation and power of GWAS.** As an approach to quantitatively evaluating the interethnic differences in blood pressure GWAS results, we estimated the genetic correlation using summary statistics of the entire spectrum of GWAS associations[18]. We first estimated the SNP-based heritability ($h^2$) of SBP and DBP (Fig. 2). For SBP, $h^2$ estimates in our study were 0.107 (SE 0.007) for East Asians and 0.086 (SE 0.009) for Europeans and lower than a previously reported UK Biobank estimate of 0.156 (SE 0.004)[22] calculated by the moment-matching method in Europeans. This discrepancy was likely due to the methodological differences in SNP-based heritability analyses between the studies but does not appear to affect genetic-correlation estimates themselves[23]. Also, the $h^2$ of DBP was almost comparable between the ethnic groups in this study. Then, we found that the genetic correlations in SBP and DBP were 0.898 (SE 0.040) and 0.851 (SE 0.046) respectively, and significantly different from 1 ($P = 0.005$ for SBP and $P = 0.0007$ for DBP). This indicated that the allele-substitution effect-sizes differed significantly between the two ethnic groups despite the reportedly substantial genetic overlap in blood pressure traits (Supplementary Data 4).

To estimate the degree of interethnic overlap and nonoverlap of blood pressure loci, we further calculated the power of GWAS of different sample sizes (i.e., 100K, 200K, and 500K) based on heritability parameters (see details in Supplementary Methods) via modeling, computing and random sampling (Fig. 3 and Supplementary Figs. 12, 13). Similar to Europeans, the recent progresses of GWAS in East Asians prompted us to investigate different sample sizes in preparation for much-larger transethnic meta-analysis. When GWASs of the same size were carried out for SBP and DBP, it was expected that an almost equivalent number of genome-wide significant loci could be identified in both East Asians and Europeans but the number of overlap was less than half.

We extended the interethnic analyses to other complex traits such as plasma lipid level, anthropometric measurement, and type 2 diabetes using published GWAS summary statistics of relatively large number of samples (Supplementary Table 6). Although genetic correlation appeared to be varied among the

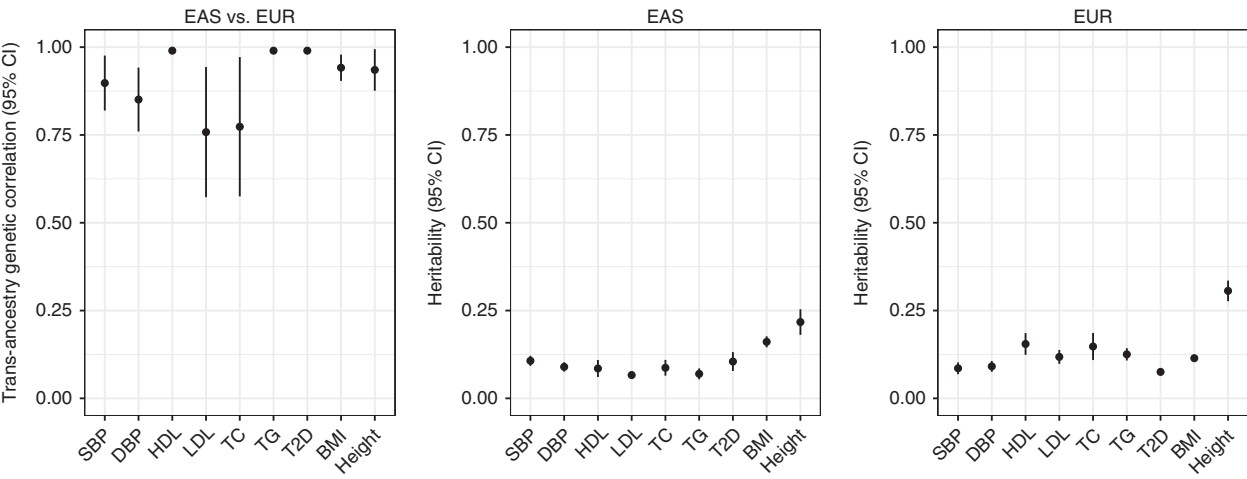

**Fig. 2** Transethnic genetic correlation and SNP-based heritability. SNP-based heritability of SBP, DBP and other complex disease and phenotype traits is shown separately for East Asians (EAS) and Europeans (EUR) by using the published GWAS summary statistics (Supplementary Table 6). The whiskers are 95% confidence intervals of each value

complex traits examined (Fig. 2), we found that the proportion of nonoverlap [(nonoverlap) / (overlap + nonoverlap)] was relatively consistent across the traits for the same sample size; 0.71–0.82 for 100K, 0.65–0.78 for 200K and 0.46–0.70 for 500K (Fig. 3 and Supplementary Fig. 12). As the sample sizes in both ethnic groups become larger, we can expect a higher proportion of interethnic overlap; nevertheless, more than or nearly half of the genome-wide significant loci may not overlap between the ethnic groups for GWAS of the same sample size.

**Selective sweeps at ancestry-specific loci.** Subsequently, we created a list of ancestry-specific loci for SBP, DBP and other complex traits in which an SNP-trait association was genome-wide significant in one ethnic group (e.g., East Asians) but no significant association signal was detectable in another (e.g., Europeans) due to low allele frequency (MAF < 0.05) (Supplementary Data 9). For the loci with the same SNPs being monomorphic in the second ethnic group, our selection criteria for ancestry-specific loci could be regarded stringent in that the absence of locus-wide significant association signals in the vicinity (≤500 kb) of the tested SNPs was required.

A larger number of significant loci had been reported in Europeans compared to East Asians, reflecting the differences in sample size of GWAS conducted to date (mean for five traits was 81,991 in East Asians vs. 202,390 in Europeans) (Fig. 4 and Supplementary Fig. 14). Thus, the total number of ancestry-specific loci across the examined traits was smaller in East Asians (10 loci) than in Europeans (63 loci). While it was most prominent for height, the sentinel SNPs at the ancestry-specific loci tended to have both lower MAF (0.20 ± 0.04 in East Asians, 0.16 ± 0.01 in Europeans) and genetic impact (0.020 ± 0.002 in East Asians, 0.014 ± 0.0004 in Europeans) across the traits.

Among a list of ancestry-specific loci for multiple traits, we identified evidence of a positive selection at five unique loci using a highly sensitive algorithm, haploPS[24] (Fig. 5 and Supplementary Data 9). For blood pressure, a sentinel SNP rs56174355 on 17q23 previously reported to be associated with DBP only in Europeans[8] was localized to a region with evidence of positive selection in East Asians (Fig. 5d). In this region, we observed the long haplotypes at high frequencies (i.e., 70–80%) to be selected exclusively in East Asians, on which the present-day major allele (G of rs56174355) could reside, whereas the minor allele (T of rs56174355) was associated with lower DBP in Europeans. Thus,

a selective sweep in the region is considered to have retained the major allele that was likely beneficial in the populations of East Asian ancestry; conversely, this has reduced MAF in East Asians (T allele: 0.03 in East Asians vs. 0.10 in Europeans). We found similar examples for the traits other than blood pressure in four regions: rs12748152 for LDL-C and triglycerides, rs17031005 for T2D, rs11862222 for height and rs4253772 for total cholesterol (Fig. 5a–c, e). There was a significant ($P_{Binomial} = 4.2 \times 10^{-5}$) increase in the incidence of recent selection signals at the ancestry-specific loci, given that a total of 405 distinct genomic regions were identified to show evidence of positive selection across 14 populations worldwide[24].

## Discussion

Our GWAS in 183,785 East Asian individuals identified 15 new genetic loci influencing blood pressure phenotypes and 4 additional loci when combined with European individuals (max $N = 289,038$) (Table 1). Of the 19 newly identified loci, 15 loci were replicated in an independent sample of Europeans ($N = 422,771$) plus East Asians ($N = 94,201$) (Supplementary Data 2). A notable feature of this study is the use of a relatively large discovery-stage sample size in populations of non-European descent, thereby enabling us to identify a number of genetic loci that have not been reported by GWAS meta-analysis in Europeans (Fig. 3). By combining the East Asian data with European data, we were also able to seek interethnic genetic heterogeneity of GWAS results for blood pressure between the two ancestries (Fig. 1) as well as other complex traits. In particular, the present study provides examples for interethnic genetic heterogeneity, although the incidence may not be high, discovering two remarkable phenomena: (1) the colocalization of distinct ancestry-specific variants that are not rare and can exert mutually inverted genetic effects between the ethnic groups and (2) the potential involvement of natural selection in the occurrence of ancestry-specific association signals.

Among genetic loci identified in East Asians, of note is the finding that at two loci on 1p35 and 3p21, the latter of which resides near the association signal previously reported in Chinese[25], sentinel SNPs (rs2076460 and rs3774447) appear to be specific to East Asians; i.e., in Europeans the corresponding SNPs were monomorphic and no significant association signals were detectable in the vicinity (Supplementary Data 7). These support the possible presence of multiple East Asian-specific associations as well as European-specific ones.

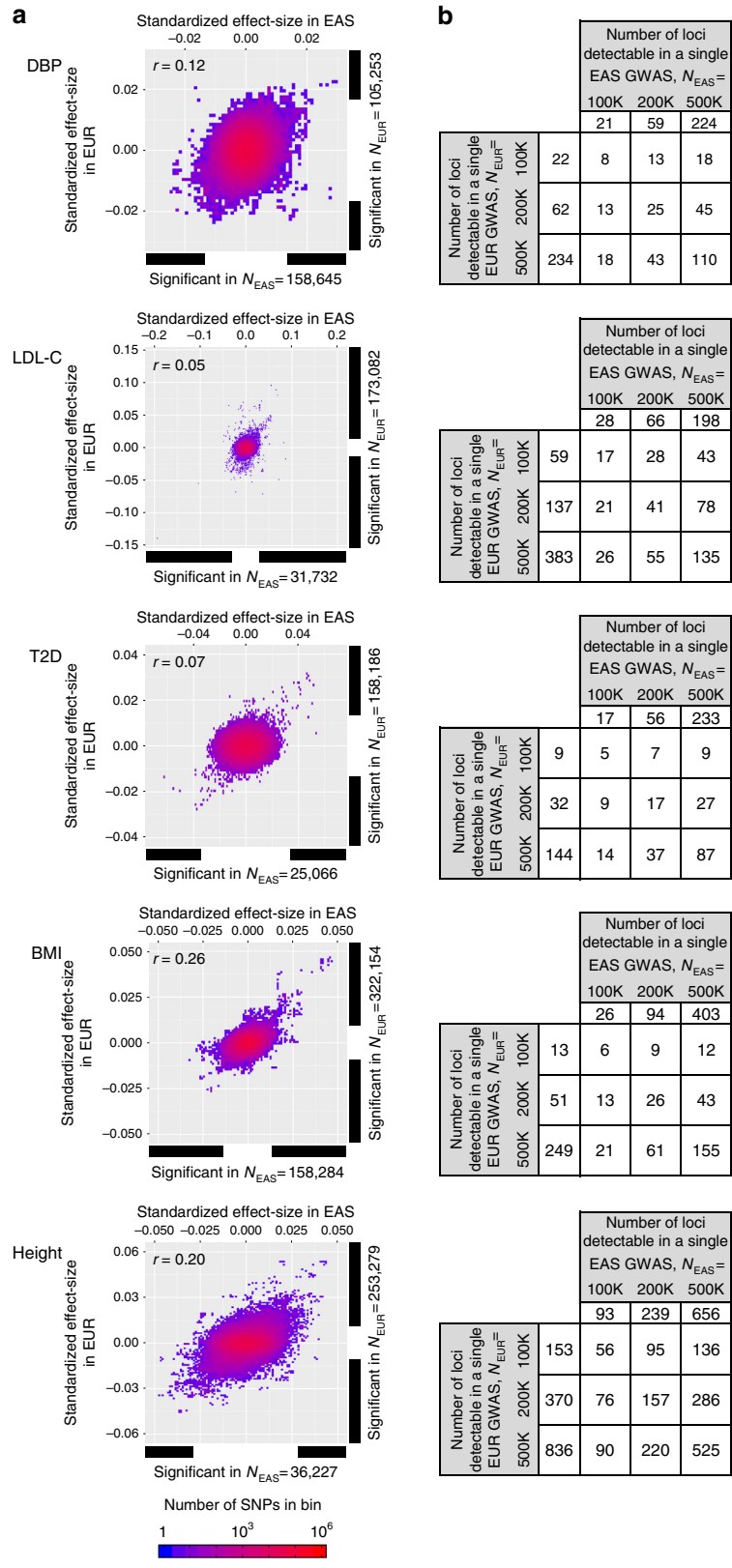

It has been suggested that GWAS signals are produced by causal variants that are common and shared between ancestry groups[26], with evidence for alternative rare variant association models (e.g., synthetic association[27]) that are assumed to be restricted to a limited number of loci. Apart from these models, we have discovered a new model in which genetic effects for transethnic SNPs that form a shared haplotype at a locus are driven by causal variants that are ancestry-specific but are not rare, which can be called a common ancestry-specific variant association model. We previously reported on 12q24 the East Asian-specific association signal at *ALDH2* with blood pressure, which was located near the association signal at *SH2B3* identified

**Fig. 3** Distribution of SNP effect-size in GWAS and power of GWAS. They are compared between East Asians and Europeans for DBP, low-density lipoprotein cholesterol (LDL-C), type 2 diabetes (T2D), body mass index (BMI) and height. **a** Distribution of SNP effect-size in actual GWAS conducted in East Asians (x-axis) and Europeans (y-axis). The effect-size of an SNP was standardized such that each of the trait and allele has a unit variance. The standardized effect-size equals the genetic impact. A positive effect-size indicates a higher trait value for the ALT allele compared to the REF allele of the 1000 Genomes (1000G) phase-3 data set. The horizontal and vertical bars to the bottom and right of the plots indicate the range of effect-sizes, in which genome-wide significant SNPs are localized. **b** The expected numbers of genome-wide significant loci detectable in a single GWAS and their interethnic overlap. The number of SNPs was scaled to 1000G SNPs even for GWAS in which HapMap-derived SNPs were assayed. SNPs located ≤500 kb were regarded to be at the same susceptibility locus. The numbers of loci were inferred from the heritability model shown in Supplementary Fig. 13, where true observable effect-sizes were computed based on 100 trials of random sampling under the assumed heritability parameters (see Methods)

in Europeans[19]. We also reported that these two association signals were phylogenetically independent, although a distance between the sentinel SNPs (rs671 and rs3184504) was relatively close (357 kb apart). Moreover, in the present study we have detected a number of transethnic SNPs in the 12q24 region to show highly significant heterogeneity of genetic impact on SBP between the ethnic groups (e.g., $\beta_{EAS} = -0.73$ and $\beta_{EUR} = 0.37$, $P_{het} = 9.74 \times 10^{-21}$ at rs4766566), where inverted genetic effects are attributable to each of the ancestry-specific sentinel SNPs; a similar situation was also observed on 10q21 to reproduce this phenomenon (Fig. 1 and Supplementary Data 6). Using ancestry-specific SNPs that are reported to reach genome-wide significance in either of the ethnic groups, we have found further evidence supporting the common ancestry-specific variant association model at 11 of 48 loci (23%) examined (Supplementary Data 7). This corresponds to 2.5% of total non-rare genome-wide significant blood pressure loci reported to date in populations of European and/or East Asian descent[3–8,19] (Supplementary Fig. 7). Although it is beyond the scope of this study, part of the low-frequency variants at group-2b loci (which constitute 4.0% of the tested blood pressure loci) may also be ancestry-specific[20]. These findings are important and should be kept in mind in the two well-known applications of transethnic GWAS, i.e., meta-analyses to increase the power for detecting new susceptibility loci and fine mapping.

With the increase in sample size used for GWAS meta-analysis, it is expected that a larger number of genetic loci will be detected, and the distribution of such loci in the genome will become denser. Association signals annotated to the same locus are empirically defined such that a set of SNPs are bounded by pairwise correlation with the index SNP of $r^2 \geq 0.1-0.3$ within ±250–500 kb of the index SNP[26,28]. This is usually discussed in the context of locus heterogeneity rather than allelic heterogeneity. Apart from extreme cases in which an index SNP is monomorphic in the second ethnic group as above-mentioned, the cases in which a common variant in question is less common or even rare in the second ethnic group necessitate a greater sample size to achieve comparable statistical power for detecting a significant association. We should be careful in setting appropriate significance thresholds to maintain a balance between generating spurious associations and missing true modest associations in the second ethnic group. Hence, we chose ancestry-specific loci based on the P value thresholds adjusted for the number of SNPs located ≤500 kb from the sentinel SNP. This set of loci may not exclude some cases with insufficient statistical power but can include the cases in which genetic impact at the locus is largely regarded as specific to the original ethnic group. Evidence of positive selection was observed at five unique loci among the list of ancestry-specific loci (Fig. 5 and Supplementary Data 9).

In addition to ancestry-specific loci, although the proportion appears to be relatively modest (approximately 2.5%), we have found significant interethnic heterogeneity of genetic impact at a number of blood pressure loci that are non-rare in both

ancestries, with most of them originally reported in Europeans to date. It is assumed that the potential presence of modifier genes and/or gene−environment interactions can contribute to such interethnic heterogeneity but the overall influences and underlying mechanisms remain to be investigated. When combined with ancestry-specific variant associations (at group-2a or group-2b loci in Supplementary Fig. 7), >5% of blood pressure loci are likely to show significant interethnic heterogeneity between East Asians and Europeans.

According to our SNP-based heritability analysis, the genome-wide correlation of causal-variant effect-sizes at SNPs common in both ancestry groups is 0.898 and 0.851 for SBP and DBP, respectively (Fig. 2). Part of the reduced interethnic correlation is attributable to transethnic variants that are common across populations but show substantial interethnic heterogeneity, although the proportion of such variants may not be high (e.g., 9 loci with interethnic heterogeneity detected in group-1; Supplementary Fig. 7). Even though they are not included in the SNP-based heritability analysis, ancestry-specific variants (at group-2a loci in Supplementary Fig. 7) can influence the per-allele effect-sizes for a number of transethnic SNPs at the corresponding loci via LD, e.g., at the C10orf107 and CACNB2 loci.

In summary, we identify a total of 19 genetic loci that have not been reported previously by GWAS meta-analysis, using relatively large discovery-stage sample size in East Asian populations. By comparing GWAS data for two ethnic groups, we have newly defined, so to speak, a common ancestry-specific variant association model, which should be brought to attention in the applications of transethnic GWAS.

## Methods

**Populations and genotyping.** Description of the study design and phenotype measurement for each East Asian study (or cohort) participating in GWAS meta-analysis is provided in the Supplementary Methods. Descriptive statistics of the individuals, genotyping arrays, quality control filters, and genotype imputation applied to the individual studies are provided in Supplementary Tables 1, 2; 1000 Genomes Phase 3 reference panel was used for imputation in all studies except BBJ (1000 Genomes Phase 1) and TMM CommCohort Study (ToMMo 2KJPN panel plus 1000 Genomes Phase 3). SNP alleles were oriented to the forward strand of the GRCh37/hg19 reference sequence of the human genome. Collection of data and samples by the cohorts participating in the study was approved by respective research ethics committees, and written consent for participation was provided by all research participants.

**Phenotype modeling and SNP association analysis.** For individuals taking antihypertensive therapies, blood pressure was imputed by adding 15 mmHg and 10 mmHg to SBP and DBP values, respectively. MAP and PP were calculated as MAP = (2 DBP + SBP)/3 and PP = SBP – DBP. In each study, the association of blood pressure (SBP, DBP, MAP or PP) with SNP allele dose was tested using linear regression adjusted for age, sex, and any study-specific covariates. Hypertensive cases were defined as follows: (i) SBP ≥ 160 mmHg and/or DBP ≥ 100 mmHg and/or on antihypertensive treatment and (ii) age of onset ≤65 years. Normotensive controls were defined as follows: (i) SBP < 130 mmHg and DBP < 85 mmHg and not on antihypertensive treatment and (ii) age ≥50 years. In each study, the association of a dichotomous trait of hypertension status with SNP allele dose was tested using logistic regression adjusted for sex and any study-specific covariates. The effect-sizes and standard errors estimated in linear and logistic regressions were used in subsequent meta-analysis.

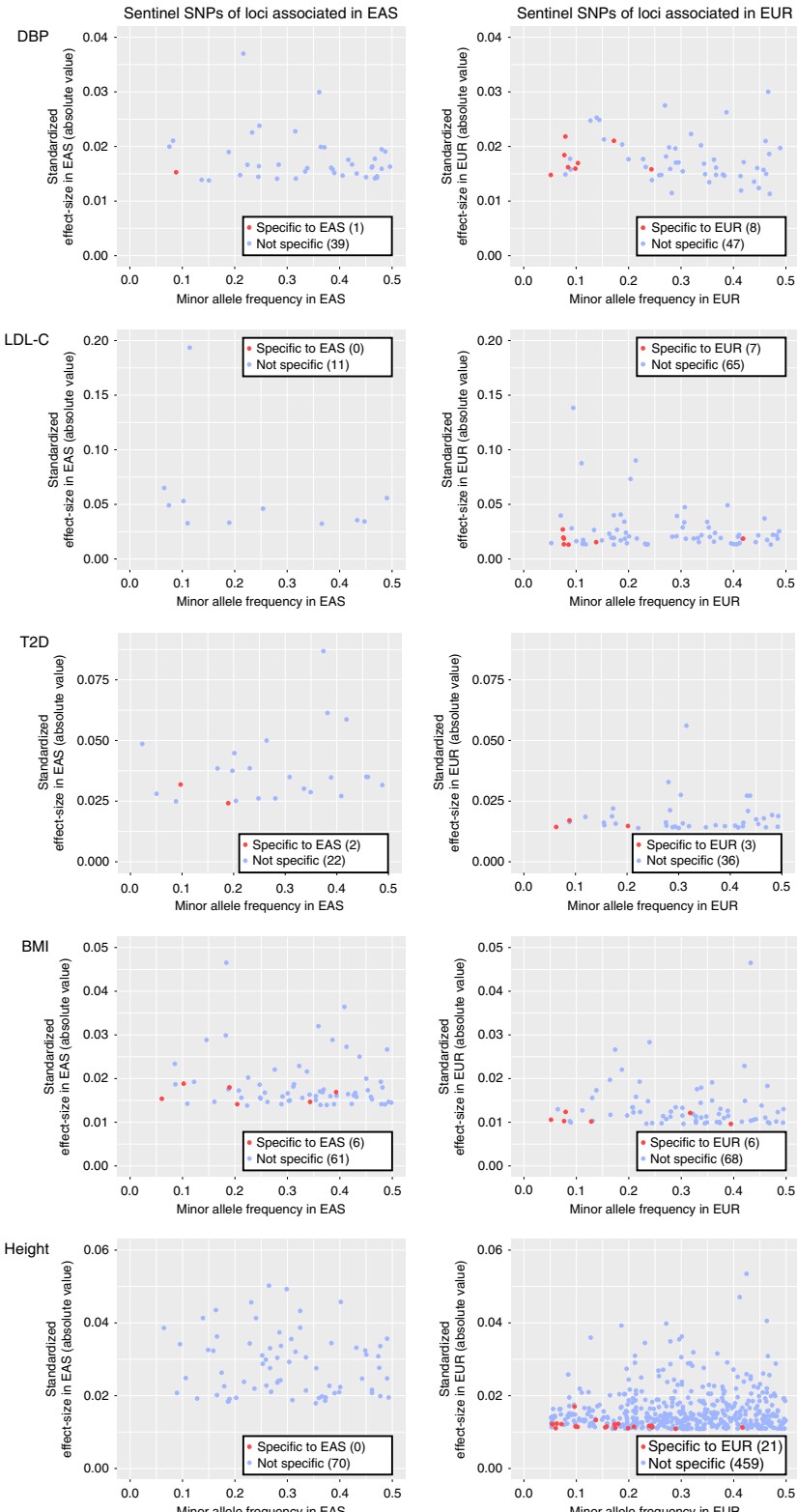

**Fig. 4** Interethnic compatibility of GWAS results for DBP, LDL-C, T2D, BMI, and height. Each point in the plots represents a sentinel SNP with genome-wide significance in the GWAS summary statistics (Supplementary Table 6), plotted with its standardized effect-size (in y-axis) against minor allele frequency (in x-axis) for East Asians (EAS in the left column) and Europeans (EUR in the right column). SNPs specific to either of the ethnic groups are colored in red; ancestry-specific association was defined such that the sentinel SNPs at the corresponding loci reached genome-wide significance ($P < 5 \times 10^{-8}$) in one ethnic group but were non-polymorphic or rare (MAF < 0.05) in another ethnic group

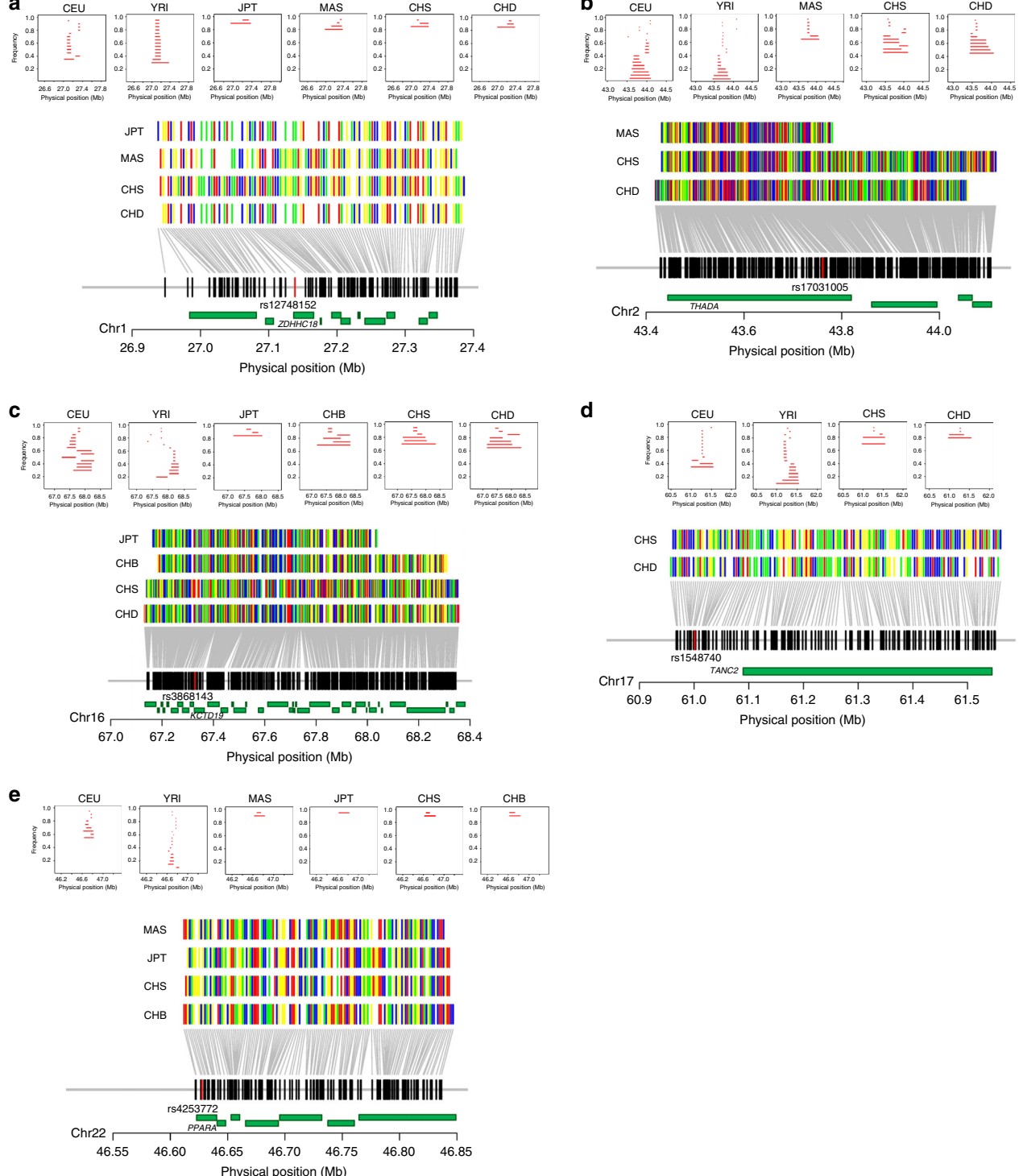

**Fig. 5** Examples of positive selection in East Asians. Selected haplotype forms are shown at five loci positively selected in East Asians. The five loci are near the following SNPs (or genes): **a** rs12748152 (*ZDHHC18*), **b** rs17031005 (*THADA*), **c** rs3868143 (*KCTD19*), **d** rs1548740 (*TANC2*) and **e** rs4253772 (*PPARA*). Selected haplotypes were identified by haploPS[24] at five sentinel SNPs out of 63 ancestry-specific loci that were identified for complex traits. In the five chromosomal regions each containing the SNP (or locus) of interest, haploPS analyses were performed across a range of core haplotype frequencies from 5 to 95%, with a frequency step size of 5%, in East Asians (including JPT, MAS, CHB, CHS and CHD) as well as Europeans (CEU) and Nigerians in West Africa (YRI) of the HapMap Phase III populations. This yielded the longest haplotype exclusively in East Asians and provided an estimate for the selected allele in its respective population, as shown in the top of each panel. For each locus, haploPS additionally located on the haplotype form on which the advantageous allele is likely to reside; each nucleotide was colored differently, adenine in green, cytosine in blue, guanine in yellow and thymine in red. In each panel, the red vertical bar indicates the position of target SNP, and gene locations (green horizontal bars) are superimposed at the bottom. At two loci, proxy SNPs in complete LD ($r^2 = 1.00$ in EAS) with the sentinel SNPs were used for the analysis; rs3868143 and rs1548740 were used instead of rs11862222 and rs56174355, respectively, because of the genotype data unavailability

**Quality control**. Before meta-analysis, quality control was applied to each study. SNPs were excluded if they had study-specific call rate < 0.95, imputation quality $R^2 < 0.5$ or MAF < 0.01. If a SNP from a study did not fit the quality standards, we regarded it as missing from that study for the purpose of meta-analysis. Results for an SNP that failed to pass the quality control filters in a given study were pooled among the other contributing studies. To detect studies with inflated GWAS significance, which can be caused by confounding biases such as population stratification, we computed the genomic control lambda ($\lambda_{GC}$)[29] and the intercept of LD Score regression[13]. A study showing a score of >1.1 for both measures was regarded as inflated. Since the LD Score regression intercept was shown to be a more powerful and accurate correction factor estimate than genomic control for GWAS with large sample size[13], we used the LD Score regression intercept as a correction factor for GWAS with a sample size of >3000 (BBJ in this study). Otherwise, $\lambda_{GC}$ was used as a correction factor.

**GWAS and replication meta-analyses**. Genome-wide association and replication studies were carried out in the multistage approach. The discovery stage (stage 1) of GWAS was carried out in a total of 130,777 East Asian individuals from five studies (Supplementary Table 1). The association results of each SNP across the studies were combined within METAL software[30] using the fixed-effects inverse-variance-weighted method. Heterogeneity of effect-sizes was tested using Cochran's Q statistic. For the stage-1 of GWAS, there were 6.2 million SNPs with heterogeneity $P > 10^{-6}$ and the sample size being at least half of the total. The Q−Q plots and Manhattan plots are shown in Supplementary Figs. 2, 3.

In the follow-up stage (stage 2) of GWAS, we considered for follow-up any SNPs with $P < 1.6 \times 10^{-5}$ for any of the five blood pressure traits. We used two follow-up stages for the East Asian-specific analyses and transethnic meta-analysis (Supplementary Fig. 1). First, we recruited additional East Asian cohorts with 1000 Genomes data and the Tohoku Medical Megabank cohort, all of which had not contributed to the GWAS stage-1 meta-analysis (max N = 53,008). Second, we sought further replication from two European GWAS data sets: the International Consortium of Blood Pressure (ICBP) (max N = 69,909)[3] and the International Genomics of Blood Pressure (iGEN-BP) Consortium (N = 35,344)[10]. This gave a total of N = 158,261 independent follow-up samples for the GWAS analysis. Combined meta-analyses of stages 1+2 data were carried out for East Asians alone (N = 183,785) as well as across the two ancestral population groups (N = 289,038). We used $P < 5 \times 10^{-8}$ to denote genome-wide significance in the combined (stages 1+2) meta-analyses. Additionally, the sentinel SNPs with $P < 5 \times 10^{-8}$ were subjected to lookups in European plus East Asian samples, including large-scale data sets for blood pressure (SBP and DBP) GWAS of the UK Biobank (N = 422,771)[14], which are publicly available via https://doi.org/10.1038/s41588-018-0144-6, and the China Kadoorie Biobank (N = 94,201)[15].

In the present study, an association signal was declared to be validated if it satisfied all four of the following criteria: (i) the sentinel SNP was genome-wide significant ($P < 5 \times 10^{-8}$) in the combined meta-analysis (stages 1+2) for any of the five blood pressure traits; (ii) the sentinel SNP showed evidence of support ($P < 0.05$) in the GWAS stage-2 alone for association with the most significantly associated blood pressure trait from the combined meta-analysis; (iii) the sentinel SNP showed further evidence of support ($P < 0.00263 = 0.05/19$) in association results for either SBP or DBP of lookup variants (n = 19 in this study); and (iv) the sentinel SNP had concordant directions of effect across the discovery and replication stages.

**Nomination of novel loci**. We reported novel loci in a unified way across the blood pressure traits. For each trait, we listed SNPs reaching genome-wide significance and filtered them by regarding two SNPs at most 500 kb apart to belong to the same locus. A blood pressure locus was defined as a chromosomal region, where a group of significant SNPs are localized ≤500 kb to the adjacent ones. For each locus, the SNP with the lowest P value was selected as a trait-specific sentinel SNP. Across the traits, all sentinel SNPs were annotated to distinct loci according to the SNP-to-SNP distance of >500 kb. Moreover, the SNP with the lowest P value across the traits was selected as a cross-trait sentinel SNP at that locus. We nominated novel loci when such cross-trait sentinel SNPs were >500 kb and not in LD ($r^2 < 0.1$ in East Asians of 1000 Genomes samples) from previously reported blood pressure SNPs at the time of analysis.

**Functional annotations and candidate gene identification**. To prioritize associated SNPs at the novel loci, we took a series of bioinformatics approaches in order to collate functional annotation (Supplementary Tables 3–5 and Supplementary Data 5). We first evaluated the sentinel SNPs for mediation of eQTLs in 14 tissues (such as the adrenal gland, artery, heart, and hypothalamus) that were considered relevant to blood pressure regulation using the GTEx v7 database[31]. We evaluated top genetic variants (eVariants) in LD ($r^2 > 0.8$) with the sentinel SNPs for evidence of mediation of eQTLs in 14 tissues using the GTEx database, to identify loci that are highly expressed and highlight specific tissue types that show eQTLs for a large proportion of the loci. Other annotations were applied to all SNPs in LD ($r^2 > 0.8$ in East Asians) with the sentinel SNPs. We used the SNPnexus[32] to provide an aggregate set of functional annotations for the SNPs, including gene location, conservation, amino acid substitution impact based on prediction tools, SIFT and PolyPhen. Previously reported association signals with other traits were looked up in the GWAS Catalogue (https://www.ebi.ac.uk/gwas/). We thus identified a list of

candidate genes at the 19 novel loci, to which ≥1 line(s) of evidence (eQTL, nonsynonymous SNP or SNP-gene colocalization) could indicate a biological link of the blood pressure SNPs.

**Interethnic heterogeneity of blood pressure GWAS results**. To examine whether genetic variants have the same phenotypic effects in different populations, we used the method for estimating the transethnic genetic correlation[18]. Briefly, in the case where two GWASs conducted on the same phenotype (i.e., blood pressure in this study) in different populations, we can consider both the correlation of allele effect-sizes and the correlation of allelic impact. The latter is defined as (per-allele effect-sizes, $\beta$) × sqrt(allele variances, $\sigma^2$), where $\sigma^2 = 2 \times$ MAF × (1 − MAF). The genetic impact at non-rare (MAF ≥ 0.01) SNPs were compared between two populations of different ancestries using GWAS data sets available in this study: East Asian samples (N = 158,645) and European samples (N = 105,253). Heterogeneity of genetic impact was tested using Cochran's Q statistic. We used genome-wide significance $P < 5 \times 10^{-8}$ to denote significant SNPs in evaluating the interethnic heterogeneity of genetic impact on SBP.

**Transethnic haplotype SNPs versus ancestry-specific SNPs**. Starting from ancestry-specific common (MAF ≥ 0.05) SNPs that were reported to reach genome-wide significance in either of the ethnic groups, we explored transethnic SNPs forming a haplotype shared between ethnic groups (denoted as haplo-SNPs) and alternate ancestry-specific SNPs in the following three steps: (i) select a sentinel SNP that was associated with blood pressure in the index ethnic group and monomorphic or MAF < 0.01 in the second ethnic group (corresponding to a group-2a SNP described below), (ii) select a haplo-SNP showing the smallest $P$ value for interethnic heterogeneity of genetic impact on a lead blood pressure trait within ±500 kb (an interval of 1 Mb) of and $r^2 \geq 0.1$ to the sentinel SNP, and (iii) select an alternate ancestry-specific SNP showing the largest genetic impact on blood pressure (i.e., the smallest $P$ value for SNP−blood pressure association) in the second ethnic group within ±500 kb of and $r^2 \geq 0.1$ to the haplo-SNP. A distance of ±500 kb and $r^2 \geq 0.1$ were set by assuming the limited recombination and LD at the locus. Interethnic differences at the haplo-SNP were considered to be significant at $P < 1.5 \times 10^{-4} \simeq 5 \times 10^{-8} \times$ [3 Gb/1 Mb] (Supplementary Data 7).

**Ancestry-specific association with complex traits**. As an approach to investigating interethnic comparability of GWAS results for complex traits, we created a list of ancestry-specific loci by using the published GWAS summary statistics (Supplementary Table 6). It was defined that at the loci, a SNP−trait association was genome-wide significant in one ethnic group (e.g., East Asians) but no association signal was detectable in another (e.g., Europeans), in which the SNP was rare (MAF < 0.05) and did not show significant association ($P > 0.05$/the number of SNPs located ≤500 kb from the sentinel SNP), considering the possible interethnic differences in genetic architecture or LD structure.

**Interethnic heterogeneity at non-rare variant loci**. We also investigated interethnic heterogeneity of genetic impact on a lead blood pressure trait at non-rare (MAF ≥ 0.01 in both ethnic groups) blood pressure loci previously reported and newly identified (Supplementary Data 8). A total of 750 previously reported SNPs (listed in Supplementary Data 4) and 19 newly identified SNPs could be classified into 485 loci by regarding two SNPs at most 500 kb apart to belong to the same locus. After exclusion of 39 loci (MAF < 0.01 in both East Asians and Europeans or no data available in GWAS data sets for both populations), 446 loci were retained and categorized into two groups—group 1 and group 2. Group 1 consisted of 382 loci with MAF ≥ 0.01 in both populations and group 2 consisted of 64 loci with potential ethnic specificity, i.e., MAF < 0.01 in either East Asians or Europeans. Group 2 was further classified into group 2a (46 loci with MAF < 0.01 in one population and MAF ≥ 0.05 in the other) and group 2b (18 loci with MAF < 0.01 in one population and 0.01 ≤ MAF < 0.05 in the other). Since ICBP and iGEN-BP (European) data were imputed with HapMap SNPs, approximately one-third of group-1 SNPs were unavailable in European GWAS data sets. Thus, 242 (out of 382) loci in group 1 were subjected to interethnic comparison of genetic impact on a lead blood pressure trait (Supplementary Fig. 7a and Supplementary Data 8). In case that there existed >1 non-rare SNPs at a locus, the SNP showing smallest $P$ value was chosen for the analysis. Also, in case that there coexisted two types— group 1 and group 2a—of SNPs at a locus, except for the *C10orf107* and *CACNB2* loci, a group-2a SNP was chosen when the remaining group-1 SNP(s) did not show significant association with blood pressure. At *C10orf107* and *CACNB2*, rs4590817 and rs12258967 (group 2a) were examined in addition to rs1530440 and rs1813353 (group 1) respectively, since the former variants were considered to be responsible for the latter association signals. Interethnic heterogeneity of genetic impact was tested using Cochran's Q statistic, where we used $P_{hetero} < 0.05/242 = 2.1 \times 10^{-4}$ to denote significant SNPs.

**SNP-based heritability analysis**. We modified the method for estimating the transethnic genetic correlation that was implemented in the Popcorn program[18] (https://github.com/brielin/popcorn). Genetic correlation measures the concordance of allele-substitution effects of causal SNPs between two populations. Popcorn is shown to use the entire spectrum of GWAS associations without raw

genotype data, while accounting for LD with the use of external reference panels (e.g., 1000 Genomes phase-3 samples) to avoid filtering correlated SNPs. Popcorn creates unbiased approximations of the genetic correlation and the population-specific heritability. We employed our method modified from Popcorn to estimate SNP-based heritability in two populations of different ancestries and to quantify transethnic genetic correlation using only summary statistics. It used to be assumed that per-SNP heritability should be equally distributed for all SNPs in a chromosomal region, but it has recently become apparent that per-SNP heritability can depend on allele frequency[33], and LD-related[34] or other functional annotations[35]. Hence we modified Popcorn by incorporating the dependence of per-SNP heritability on allele frequency and LD-related functional annotations (see details in Supplementary Methods).

We estimated SNP-based heritability of complex disease and phenotype traits including blood pressure (SBP and DBP), plasma lipid level (LDL-cholesterol, HDL-cholesterol, total cholesterol and triglycerides)[36,37], type 2 diabetes[38,39], and anthropometric measurement (BMI[40,41] and height[42,43]), for which summary statistics of relatively large ($N > 25,000$ individuals per ethnic group) GWAS meta-analysis are available for both East Asians and Europeans at the time of analysis.

**Power calculation of GWAS based on heritability parameters**. We estimated the power of a GWAS of different sample sizes (i.e., 100K, 200K, and 500K) based on heritability parameters (see details in Supplementary Methods). Briefly, we first computed the distribution of standardized effect-sizes of SNPs, which are the correlation between the SNP genotype and the phenotype and observable as the Z-statistics divided by the square root of sample size in GWAS. We modeled the effect-size distribution based on the observed heritability parameters. By iterative computing and random sampling, we could obtain one possible instance of true observable effect-size for the significant SNPs under the assumed heritability parameters. For this true effect-size, we computed the expected number of genome-wide significant SNPs (or loci) showing equal to or larger than the given value in a GWAS of a given sample size (Supplementary Fig. 13). For a pair of GWASs, we then calculated the number of overlapping genome-wide significant loci (Fig. 3 and Supplementary Fig. 12).

**Testing selection signals at ancestry-specific loci**. We tested the hypothesis that natural selection could play a role in ancestry-specific association signals of complex traits, by using the findings in the previous HaploPS analysis[24]. HaploPS is a highly sensitive algorithm to locate genomic signatures of positive selection and to allow for the detection of the founder haplotype form that carries the selected allele. HaploPS had successfully identified 405 distinct genomic regions exhibiting evidence of positive selection across 14 populations worldwide. We compared this list of 405 regions with 63 ancestry-specific loci (or the respective sentinel SNPs) identified for complex traits in search of their colocalization.

**Code availability**. The source code for SNP-based heritability analysis is publicly available (https://github.com/fumi-github/Popcorn-t).

## Data availability

Full summary statistics relating to the GWAS meta-analysis has been deposited at the European Genome-phenome Archive (EGA), which is hosted by the EBI and the CRG, under accession number EGAS00001002991. Further information about EGA can be found on https://ega-archive.org "The European Genome-phenome Archive of human data consented for biomedical research" (http://www.nature.com/ng/journal/v47/n7/full/ng.3312.html). All relevant data are available from the authors.

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

## Acknowledgements

We acknowledge the use of data from the International Consortium for Blood Pressure Genome-Wide Association Studies[3] and the AGEN-height Consortium[42]. This research has been conducted using the UK Biobank Resource[14]. Additional acknowledgements can be found in Supplementary Note 1.

## Author contributions

Participant recruitment, characterization and data generation. Anti-aging study cohort: Y.T., K.K., M. Igase, T. Miki; Biobank Japan: A.T., Y.M., M.H., K.M., M.K., Y.K.; Beijing Eye Study: Y.X.W., W.B.W., L.X., J.B.J.; CAGE_GWAS1: E.N., T.N., N.K.; CAGE-Amagasaki: T.K., H.R., M. Isono, N.K.; China Health and Nutrition Survey: W.H., K.L.M.; Cebu Longitudinal Health and Nutrition Survey: L.S.A., K.L.M.; GenSalt Study: J.H.; Korea Association Resource: B.-J.K.; Kita-Nagoya Genomic Epidemiology study: M.N., S. I., T. Matsubara, K.Y., M. Yokota; Living Biobank: D.F.R., E.-S.T.; Nutrition and Health of Aging Population in China Cohort: H. Li, L.S., X.L.; Singapore Chinese Eye Study: C.-Y.C.; Singapore Chinese Health Study: J.-M.Y., W.-P.K., Y.F., C.-K.H.; Singapore Malay Eye Study: C.S., T.Y.W.; Singapore Prospective Study Program: J. Liu, X.S.; Shanghai

Men's Health Study: Jing H., H.C., Y.-B.X., X.-O.S.; Shanghai Women's Health Study: Honglan L., T.E., J. Long, W. Zheng; TMM CommCohort Study: M.Sasaki, M.Yamamoto; Taiwan Super Control Study: J.-Y.W., Y.-T.C.; China Kadoorie Biobank: R.C.; R.G. W.; I.Y.M.; L.L.; Z.C. Statistical analyses: F.T., Y.T., M.A., N.M., Y.X.W., W.B.W., L.X., J. B.J., S. Huo, C.N.S., P.G.-L., W.H., K.L.M., N.R.L., L.S.A., Y.-J.K., M.Y.H., J. Lee, S.M., S. Han, M.N., C.W., S. Huo, W.-Y.S., Y.-Y.T., W. Zhao, R.D., C.-K.H., L.Z., M.L.C., J.-F.C., L.-C.C., C.-H.C., A.H., A.N., M. Shirota, J.L.N., T.N.K., D.A.B., K.L. (with group lead, F. T.). Manuscript writing: N.K., F.T., W.-Y.S., Y.-Y.T., X.S., C.N.S., K.L.M. (with group lead, N.K.). All authors critically reviewed and approved the final version of the manuscript.

## Additional information

**Competing interests:** The authors declare no competing interests.

Fumihiko Takeuchi[1,2], Masato Akiyama[3], Nana Matoba[3], Tomohiro Katsuya[4,5], Masahiro Nakatochi[6], Yasuharu Tabara[7], Akira Narita[8], Woei-Yuh Saw[9,10], Sanghoon Moon[11], Cassandra N. Spracklen[12], Jin-Fang Chai[9], Young-Jin Kim[11], Liang Zhang[13], Chaolong Wang[14,15,16], Huaixing Li[17], Honglan Li[18], Jer-Yuarn Wu[19,20], Rajkumar Dorajoo[14], Jovia L. Nierenberg[21], Ya Xing Wang[22], Jing He[23], Derrick A. Bennett[24], Atsushi Takahashi[3,25], Yukihide Momozawa[26], Makoto Hirata[27], Koichi Matsuda[28], Hiromi Rakugi[5], Eitaro Nakashima[29,30], Masato Isono[2], Matsuyuki Shirota[8], Atsushi Hozawa[8], Sahoko Ichihara[31], Tatsuaki Matsubara[32], Ken Yamamoto[33], Katsuhiko Kohara[34], Michiya Igase[35], Sohee Han[11], Penny Gordon-Larsen[36,37], Wei Huang[38], Nanette R. Lee[39,40], Linda S. Adair[36,37], Mi Yeong Hwang[11], Juyoung Lee[11], Miao Li Chee[13], Charumathi Sabanayagam[13,41,42], Wanting Zhao[13,15], Jianjun Liu[14,43], Dermot F. Reilly[44], Liang Sun[17], Shaofeng Huo[17], Todd L. Edwards[23], Jirong Long[23], Li-Ching Chang[19], Chien-Hsiun Chen[19], Jian-Min Yuan[45,46], Woon-Puay Koh[9,47], Yechiel Friedlander[48], Tanika N. Kelly[21], Wen Bin Wei[49], Liang Xu[22], Hui Cai[23], Yong-Bing Xiang[18], Kuang Lin[24], Robert Clarke[24], Robin G. Walters[24], Iona Y. Millwood[24,50], Liming Li[51,52], John C. Chambers[53], Jaspal S. Kooner[54], Paul Elliott[55,56,57,58,59], Pim van der Harst[60], The International Genomics of Blood Pressure (iGEN-BP) Consortium, Zhengming Chen[24], Makoto Sasaki[61], Xiao-Ou Shu[23], Jost B. Jonas[22,62], Jiang He[21,63], Chew-Kiat Heng[64,65], Yuan-Tsong Chen[19], Wei Zheng[23], Xu Lin[17], Yik-Ying Teo[9,10,66],

E-Shyong Tai[9,15,43], Ching-Yu Cheng[13,41,42], Tien Yin Wong[13,41,42], Xueling Sim [9], Karen L. Mohlke[12], Masayuki Yamamoto[8], Bong-Jo Kim[11], Tetsuro Miki[35], Toru Nabika[67], Mitsuhiro Yokota[68], Yoichiro Kamatani [3,69], Michiaki Kubo[70] & Norihiro Kato[1,2]

[1]Medical Genomics Center, National Center for Global Health and Medicine, Tokyo 162-8655, Japan. [2]Department of Gene Diagnostics and Therapeutics, Research Institute, National Center for Global Health and Medicine, Tokyo 162-8655, Japan. [3]Laboratory for Statistical Analysis, RIKEN Center for Integrative Medical Sciences, Yokohama 230-0045, Japan. [4]Department of Clinical Gene Therapy, Osaka University Graduate School of Medicine, Suita 565-0871, Japan. [5]Department of Geriatric and General Medicine, Osaka University Graduate School of Medicine, Suita 565-0871, Japan. [6]Data Coordinating Center, Department of Advanced Medicine, Nagoya University Hospital, Nagoya 466-8560, Japan. [7]Center for Genomic Medicine, Kyoto University Graduate School of Medicine, Kyoto 606-8507, Japan. [8]Tohoku Medical Megabank Organization, Tohoku University, Sendai 980-8573, Japan. [9]Saw Swee Hock School of Public Health, National University of Singapore and National University Health System, Singapore 117549, Singapore. [10]Life Sciences Institute, National University of Singapore, Singapore 117456, Singapore. [11]Division of Genome Research, Center for Genome Science, National Institute of Health, Chungcheongbuk-do 363-951, Republic of Korea. [12]Department of Genetics, University of North Carolina, Chapel Hill NC 27514, USA. [13]Singapore Eye Research Institute, Singapore National Eye Centre, Singapore 168751, Singapore. [14]Genome Institute of Singapore, Agency for Science, Technology and Research, Singapore 138672, Singapore. [15]Duke-NUS Medical School, Singapore 169857, Singapore. [16]Department of Epidemiology and Biostatistics, School of Public Health, Tongji Medical College, Huazhong University of Science and Technology, Wuhan 430030, China. [17]CAS Key Laboratory of Nutrition, Metabolism and Food safety, Shanghai Institute of Nutrition and Health, Shanghai Institutes for Biological Sciences, University of the Chinese Academy of Sciences, Chinese Academy of Sciences, Shanghai 200031, China. [18]State Key Laboratory of Oncogene and Related Genes and Department of Epidemiology, Shanghai Cancer Institute, Renji Hospital, Shanghai Jiaotong University School of Medicine, Shanghai 200025, China. [19]Institute of Biomedical Sciences, Academia Sinica, Taipei 115, Taiwan. [20]School of Chinese Medicine, China Medical University, Taichung 40402, Taiwan. [21]Department of Epidemiology, Tulane University School of Public Health and Tropical Medicine, New Orleans LA 70112, USA. [22]Beijing Institute of Ophthalmology, Beijing Key Laboratory of Ophthalmology and Visual Sciences, Beijing Tongren Eye Center, Beijing Tongren Hospital, Capital Medical University, Beijing 100730, China. [23]Division of Epidemiology, Vanderbilt University Medical Center, Nashville TN 37203-1738, USA. [24]Clinical Trial Service Unit and Epidemiological Studies Unit, Nuffield Department of Population Health, University of Oxford, Oxford OX3 7LF, UK. [25]Department of Genomic Medicine, Research Institute, National Cerebral and Cardiovascular Center, Osaka 565-0873, Japan. [26]Laboratory for Genotyping Development, RIKEN Center for Integrative Medical Sciences, Yokohama 230-0045, Japan. [27]Institute of Medical Science, The University of Tokyo, Tokyo 108-8639, Japan. [28]Graduate School of Frontier Sciences, The University of Tokyo, Kashiwa 277-8561, Japan. [29]Division of Endocrinology and Diabetes, Department of Internal Medicine, Nagoya University Graduate School of Medicine, Nagoya 466-8550, Japan. [30]Department of Diabetes and Endocrinology, Chubu Rosai Hospital, Nagoya 455-8530, Japan. [31]Department of Environmental and Preventive Medicine, Jichi Medical University School of Medicine, Shimotsuke 329-0498, Japan. [32]Department of Internal Medicine, School of Dentistry, Aichi Gakuin University, Nagoya 470-0195, Japan. [33]Department of Medical Biochemistry, Kurume University School of Medicine, Kurume 830-0011, Japan. [34]Faculty of Collaborative Regional Innovation, Ehime University, Matsuyama 790-8577 Ehime, Japan. [35]Department of Geriatric Medicine, Ehime University Graduate School of Medicine, Toon 791-0295 Ehime, Japan. [36]Department of Nutrition, Gillings School of Global Public Health, University of North Carolina at Chapel Hill, Chapel Hill NC 27599, USA. [37]Carolina Population Center, University of North Carolina at Chapel Hill, Chapel Hill NC 27516, USA. [38]Department of Genetics, Shanghai-MOST Key Laboratory of Health and Disease Genomics, Chinese National Human Genome Center and Shanghai Industrial Technology Institute (SITI), Shanghai 201203, China. [39]USC-Office of Population Studies Foundation, University of San Carlos, Cebu City 6000, Philippines. [40]Department of Anthropology, Sociology and History, University of San Carlos, Cebu City 6000, Philippines. [41]Ophthalmology & Visual Sciences Academic Clinical Program (Eye ACP), Duke-NUS Medical School, Singapore 169857, Singapore. [42]Department of Ophthalmology, Yong Loo Lin School of Medicine, National University of Singapore, Singapore 119228, Singapore. [43]Yong Loo Lin School of Medicine, National University of Singapore, Singapore 117597, Singapore. [44]Merck Sharp Dohme Corp, Kenilworth NJ 07033, USA. [45]Department of Epidemiology, Graduate School of Public Health, University of Pittsburgh, Pittsburgh PA 15261, USA. [46]Division of Cancer Control and Population Sciences, UPMC Hillman Cancer, University of Pittsburgh, Pittsburgh PA 15232, USA. [47]Health Services and Systems Research, Duke-NUS Medical School, Singapore 169857, Singapore. [48]Unit of Epidemiology, Hebrew University-Hadassah Braun School of Public Health, Jerusalem P.O. Box 12272, Israel. [49]Beijing Tongren Eye Center, Beijing Tongren Hospital, Capital Medical University, Beijing 100730, China. [50]MRC Population Health Research Unit, Nuffield Department of Population Health, University of Oxford, Oxford OX3 7LF, UK. [51]Chinese Academy of Medical Sciences, Beijing 100006, China. [52]Department of Epidemiology and Biostatistics, School of Public Health, Peking University, Beijing 100191, China. [53]Department of Epidemiology and Biostatistics, Imperial College London, London SW7 2AZ, UK. [54]National Heart and Lung Institute, Imperial College London, London SW7 2AZ, UK. [55]Medical Research Council-Public Health England (MRC-PHE) Centre for Environment and Health, Department of Epidemiology and Biostatistics, School of Public Health, Faculty of Medicine, Imperial College London, London SW7 2AZ, UK. [56]Imperial College Biomedical Research Centre, Imperial College London, London SW7 2AZ, UK. [57]UK-Dementia Research Institute at Imperial College London, London SW7 2AZ, UK. [58]National Institute for Health Research (NIHR) Health Protection Research Unit on Health Impacts of Environmental Hazards, London SW7 2AZ, UK. [59]Health Data Research UK-London, London, UK. [60]Department of Cardiology, University of Groningen, University Medical Center Groningen, Groningen 9700 RB, Netherlands. [61]Iwate Tohoku Medical Megabank Organization, Iwate 028-3694, Japan. [62]Department of Ophthalmology, Medical Faculty Mannheim, University Heidelberg, Germany, Mannheim 68167, Germany. [63]Department of Medicine, Tulane University School of Medicine, New Orleans LA 70112 LA, USA. [64]Department of Paediatrics, Yong Loo Lin School of Medicine, National University of Singapore, Singapore 119228, Singapore. [65]Khoo Teck Puat-National University Children's Medical Institute, National University Health System, Singapore 119228, Singapore. [66]NUS Graduate School for Integrative Science and Engineering, National University of Singapore, Singapore 119077, Singapore. [67]Department of Functional Pathology, Shimane University Faculty of Medicine, Izumo 693-0021, Japan. [68]Department of Genome Science, School of Dentistry, Aichi Gakuin University, Nagoya 464-8650, Japan. [69]Kyoto-McGill International Collaborative School in Genomic Medicine, Kyoto University Graduate School of Medicine, Kyoto 606-8501, Japan. [70]RIKEN Center for Integrative Medical Sciences, Yokohama 230-0045, Japan. These authors contributed equally: Fumihiko Takeuchi, Masato Akiyama, Nana Matoba, Tomohiro Katsuya, Masahiro Nakatochi, Yasuharu Tabara. These authors jointly supervised this work: Tetsuro Miki, Toru Nabika, Mitsuhiro Yokota, Yoichiro Kamatani, Michiaki Kubo, Norihiro Kato. A full list of consortium members appears at the end of the paper.

# The International Genomics of Blood Pressure (iGEN-BP) Consortium

Marie Loh[71,72], Niek Verweij[73], Weihua Zhang[72,74], Benjamin Lehne[72], Irene Mateo Leach[73], Alexander Drong[75], James Abbott[76], Sian-Tsung Tan[74,77], William R. Scott[72,77], Gianluca Campanella[72], Marc Chadeau-Hyam[72], Uzma Afzal[72,74], Tõnu Esko[78,79,80,81], Sarah E. Harris[82,83], Jaana Hartiala[84,85], Marcus E. Kleber[86], Richa Saxena[87], Alexandre F.R. Stewart[88,89], Tarunveer S. Ahluwalia[90], Imke Aits[91], Alexessander Da Silva Couto Alves[92], Shikta Das[92], Jemma C. Hopewell[93], Robert W. Koivula[94], Leo-Pekka Lyytikäinen[95,96], Iris Postmus[97,98], Olli T. Raitakari[99,100], Robert A. Scott[101], Rossella Sorice[102], Vinicius Tragante[103], Michela Traglia[104,105], Jon White[106], Inês Barroso[107,108,109], Andrew Bjonnes[87], Rory Collins[103], Gail Davies[110], Graciela Delgado[86], Pieter A. Doevendans[103], Lude Franke[111], Ron T. Gansevoort[112], Tanja B. Grammer[86], Niels Grarup[86], Jagvir Grewal[72,74], Anna-Liisa Hartikainen[113,114], Stanley L. Hazen[115,116], Chris Hsu[117], Lise L.N. Husemoen[118], Johanne M. Justesen[90], Meena Kumari[119], Wolfgang Lieb[91], David C.M. Liewald[110], Evelin Mihailov[78], Lili Milani[78], Rebecca Mills[74], Nina Mononen[95,96], Kjell Nikus[120], Teresa Nutile[102], Sarah Parish[93], Olov Rolandsson[121], Daniela Ruggiero[102], Cinzia F. Sala[104], Harold Snieder[122], Thomas H.W. Spasø[90], Wilko Spiering[123], John M. Starr[83,124], David J. Stott[125], Daniel O. Stram[117], Silke Szymczak[126], W.H.Wilson Tang[115,116], Stella Trompet[127], Väinö Turjanmaa[128,129], Marja Vaarasmaki[130], Wiek H. van Gilst[73], Dirk J. van Veldhuisen[73], Jorma S. Viikari[131,132], Folkert W. Asselbergs[103,133,134], Marina Ciullo[102], Andre Franke[126], Paul W. Franks[94,121,135], Steve Franks[136], Myron D. Gross[137], Torben Hansen[90], Marjo-Riitta Jarvelin[72,92,138,139,140], Torben Jørgensen[118], Wouter J. Jukema[127,133,141], Mika Kähönen[128,129], Mika Kivimaki[119], Terho Lehtimäki[95,96], Allan Linneberg[118], Oluf Pedersen[90], Nilesh J. Samani[142,143], Daniela Toniolo[104,144], Hooman Allayee[84,85], Ian J. Deary[83,110], Winfried März[86,145,146], Andres Metspalu[78], Cisca Wijmenga[111], Bruce H.W. Wolffenbuttel[147], Paolo Vineis[72], Soterios A. Kyrtopoulos[148], Jos C.S. Kleinjans[149], Mark I. McCarthy[75,150] & James Scott[77]

[71]Institute of Health Sciences, University of Oulu, P.O.Box 5000FI-90014 Oulu, Finland. [72]Department of Epidemiology and Biostatistics, Imperial College London, London W2 1PG, UK. [73]Department of Cardiology, University Medical Center Groningen, University of Groningen, Hanzeplein 1, 9713 GZ Groningen, Netherlands. [74]Ealing Hospital NHS Trust, Middlesex UB1 3HW, UK. [75]Wellcome Trust Centre for Human Genetics, University of Oxford, Oxford OX3 7BN, UK. [76]Bioinformatics Support Service, Imperial College London, South Kensington, London SW7 2AZ, UK. [77]National Heart and Lung Institute, Imperial College London, London W12 0NN, UK. [78]Estonian Genome Center, University of Tartu, Riia 23c, 51010 Tartu, Estonia. [79]Division of Endocrinology, Children's Hospital Boston, Longwood 300, Boston, MA 02115, USA. [80]Department of Genetics, Harvard Medical School, 77 Avenue Louis Pasteur, Boston, MA 02115, USA. [81]Program in Medical and Population Genetics, Broad Institute, 7 Cambridge Center, Cambridge, MA 02142, USA. [82]Medical Genetics Section, University of Edinburgh Molecular Medicine Centre and MRC Institute of Genetics and Molecular Medicine, Western General Hospital, Crewe Road, Edinburgh EH4 2XU, UK. [83]Centre for Cognitive Aging and Cognitive Epidemiology, University of Edinburgh, Edinburgh EH8 9JZ, UK. [84]Department of Preventive Medicine, USC Keck School of Medicine, Los Angeles, CA 90033, USA. [85]Institute for Genetic Medicine, USC Keck School of Medicine, Los Angeles, CA 90033, USA. [86]Medical Clinic V, Mannheim Medical Faculty, University of Heidelberg, Theodor-Kutzer-Ufer 1-3, 68167 Mannheim, Germany. [87]Massachusetts General Hospital, Harvard Medical School, Boston, MA 02114, USA. [88]University of Ottawa Heart Institute, Cardiovascular Research Methods Centre, Ontario K1Y 4W7, Canada. [89]Ruddy Canadian Cardiovascular Genetics Centre, Ontario K1Y 4W7, Canada. [90]Novo Nordisk Foundation Centre for Basic Metabolic Research, Section of Metabolic Genetics, Faculty of Health and Medical Sciences, University of Copenhagen, 2100 Copenhagen, Denmark. [91]Institute of Epidemiology and Biobank Popgen, Christian-Albrechts-University of Kiel, 24105 Kiel, Germany. [92]Department of Epidemiology and Biostatistics, MRC Health Protection Agency (HPE) Centre for Environment and Health, School of Public Health, Imperial College London, London SW7 2AZ, UK. [93]Clinical Trial Service Unit & Epidemiological Studies Unit, University of Oxford, Richard Doll Building, Old Road Campus, Roosevelt Drive, Oxford OX3 7LF, UK. [94]Department of Clinical Sciences, Genetic and Molecular Epidemiology Unit, Skåne University Hospital Malmö, SE-205 02 Malmö, Sweden. [95]Department of Clinical Chemistry, Fimlab Laboratories, FI-33520 Tampere, Finland. [96]Department of Clinical Chemistry, University of Tampere School of Medicine, FI-33014 Tampere, Finland. [97]Department of Gerontology and Geriatrics, Leiden University Medical Center, 2300 RC Leiden, Netherlands. [98]Netherlands Consortium for Healthy Ageing, Leiden 2333 ZC, Netherlands. [99]Department of Clinical Physiology and Nuclear Medicine, Turku University Hospital, FI-20521 Turku, Finland. [100]Research Centre of Applied and Preventive Cardiovascular Medicine, University of Turku, FI-20520 Turku, Finland. [101]MRC Epidemiology Unit, Institute of Metabolic Science, University of Cambridge, Cambridge CB2 0QQ, UK. [102]Institute of Genetics and Biophysics A. Buzzati-Traverso, CNR, 80131 Naples, Italy. [103]Department of Cardiology, Division Heart and Lungs, University Medical Center Utrecht, 3508 GA Utrecht, Netherlands. [104]Division of Genetics and Cell Biology, San Raffaele Scientific Institute, 20132 Milano, Italy. [105]Institute for Maternal and Child Health—IRCCS ''Burlo Garofolo''—Trieste, 34137 Trieste, Italy. [106]UCL Genetics Institute, Department of Genetics, Environment and Evolution, UCL, London WC1E 6BT, UK. [107]Metabolic Disease Group, The Wellcome Trust Sanger Institute, Cambridge CB10 1SA, UK. [108]NIHR Cambridge Biomedical Research Centre, Institute of Metabolic Science, Addenbrooke's Hospital, Cambridge CB2 0QQ, UK. [109]University of Cambridge Metabolic Research Laboratories, Institute of Metabolic Science, Addenbrooke's Hospital, Cambridge CB2 0QQ, UK. [110]Department of Psychology, University of Edinburgh, 7 George Square, Edinburgh EH8 9JZ, UK. [111]Department of Genetics, University Medical Center, University of Groningen, Hanzeplein 1, 9713 GZ Groningen, Netherlands. [112]Department of Internal Medicine,

University Medical Center Groningen, University of Groningen, Hanzeplein 1, 9713 GZ Groningen, Netherlands. [113]Department of Obstetrics and Gynecology, University Hospital of Oulu, University of Oulu, Oulu FI-90014, Finland. [114]Department of Clinical Sciences/Obsterics and Gynecology, University of Oulu, Oulu FI-90014, Finland. [115]Center for Cardiovascular Diagnostics and Prevention, Cleveland Clinic, Cleveland, OH 44195, USA. [116]Department of Cellular and Molecular Medicine, Lerner Research Institute, Cleveland Clinic, Cleveland, OH 44195, USA. [117]Keck School of Medicine, University of Southern California, Los Angeles, CA 90089, USA. [118]Research Centre for Prevention and Health, Glostrup University Hospital, 2600 Glostrup, Denmark. [119]Department of Epidemiology and Public Health, UCL, London WC1E 6BT, UK. [120]Heart Centre, Department of Cardiology, Tampere University Hospital, and University of Tampere School of Medicine, FI-33521 Tampere, Finland. [121]Department of Public Health & Clinical Medicine, Section for Family Medicine, Umeå universitet, SE-901 85 Umeå, Sweden. [122]Department of Epidemiology, University Medical Center Groningen, University of Groningen, Hanzeplein 1, 9713 GZ Groningen, Netherlands. [123]Department of Vascular Medicine, University Medical Center Utrecht, 3508 GA Utrecht, Netherlands. [124]Alzheimer Scotland Dementia Research Centre, University of Edinburgh, 7 George Square, Edinburgh EH8 9JZ, UK. [125]Academic Section of Geriatric Medicine, Institute of Cardiovascular and Medical Sciences, Faculty of Medicine, University of Glasgow, Glasgow G4 0SF, UK. [126]Institute of Clinical Molecular Biology, Christian-Albrechts-University of Kiel, Kiel 24105, Germany. [127]Department of Cardiology, Leiden University Medical Center, 2300 RC Leiden, Netherlands. [128]Department of Clinical Physiology, Tampere University Hospital, FI-33521 Tampere, Finland. [129]Department of Clinical Physiology, University of Tampere School of Medicine, FI-33014 Tampere, Finland. [130]Department of Obstetrics and Gynecology, Oulu University Hospital, PO Box 23FI-90029 Oulu, Finland. [131]Department of Medicine, Turku University Hospital, FI-20521 Turku, Finland. [132]Department of Medicine, University of Turku, FI-20014 Turku, Finland. [133]Durrer Center for Cardiogenetic Research, ICIN-Netherlands Heart Institute, 3511 GC Utrecht, Netherlands. [134]Institute of Cardiovascular Science, Faculty of Population Health Sciences, University College London, London WC1E 6BT, UK. [135]Department of Nutrition, Harvard School of Public Health, Boston, MA 02115, USA. [136]Institute of Reproductive and Developmental Biology, Imperial College London, Hammersmith Hospital, London W12 0HS, UK. [137]School of Medicine, University of Minnesota, Minneapolis, MN 55455, USA. [138]Biocenter Oulu, University of Oulu, P.O. Box 5000Aapistie 5A, FI-90014 Oulu, Finland. [139]Unit of Primary Care, Oulu University Hospital, Kajaanintie 50P.O.Box 20FI-90220 Oulu, Finland. [140]Department of Children and Young People and Families, National Institute for Health and Welfare, Aapistie 1, Box 310, FI-90101 Oulu, Finland. [141]Interuniversity Cardiology Institute of the Netherlands, Utrecht 3511 EP, Netherlands. [142]Department of Cardiovascular Sciences, University of Leicester, Glenfield Hospital, Leicester LE3 9QP, UK. [143]National Institute for Health Research Leicester Cardiovascular Biomedical Research Unit, Glenfield Hospital, Leicester LE3 9QP, UK. [144]Institute of Molecular GeneticsCNR, 27100 Pavia, Italy. [145]Clinical Institute of Medical and Chemical Laboratory Diagnostics, Medical University of Graz, Auenbruggerplatz 15, 8036 Graz, Austria. [146]Synlab Academy, Synlab Services GmbH, Gottlieb-Daimler-Straße 25, 68165 Mannheim, Germany. [147]Department of Endocrinology, University Medical Center Groningen, University of Groningen, Hanzeplein 1, 9713 GZ Groningen, Netherlands. [148]National Hellenic Research Foundation, Institute of Biological Research and Biotechnology, Athens 116 35, Greece. [149]Department of Toxicogenomics, Maastricht University, Universiteitssingel 50, 6229ER Maastricht, Netherlands. [150]Oxford Centre for Diabetes Endocrinology and Metabolism, University of Oxford, Oxford OX3 7LE, UK

