## [Peer Review File · Nature Communications]

Reviewer #1 (Remarks to the Author):

This paper initially describes a GWAS for BP phenotypes, expanding on this groups earlier work on BP discovery and reports 19 new BP loci. The authors then go onto describe inter-ethnic heterogeneity of the signals at some BP loci (there are some novel observations here, although one of the genome-wide significant loci was the topic of some of this groups prior work). Up until this point the paper is easy to follow and interpret the findings, although the section on ancestry-specific SNP loci should follow next. The analyses to delve into ancestry specific loci for BP and other traits is not so well described and easy to follow. The authors need to define their motivation for pursuing the additional analyses and across traits and make their observations more succinct, including the power analyses. At the moment the paper is a bit of a mix of topics, it may work better to describe all the work on BP then compare to results from other traits and then make the statements on common-ancestry specific association model.

Minor comments

1. Can the authors indicate which imputation panel was used for the discovery analyses, this is not stated in the text. I note the information is in Suppl. Table 2.
2. Sentinel variants in a combined meta-analysis of discovery and replication samples reaching $P < 5 \times 10^{-8}$ are reported as novel loci in this study – can this definition be stated in the first paragraph of the results so the study design is totally clear in the text alongside the supplementary figure.
3. ST6 presents the results from eQTL analyses, there is no text provided on the exact method and any indication of the coincidence of the BP SNP and top eQTL for the gene indicated? From a brief look at the results I do not think many of the BP SNPs are in high LD ($r^2 > 0.8$) with the top eSNP for some of the genes listed. The eQTL results should be carefully reviewed for presentation.
4. The section entitled genetic correlation and power of GWAS requires a better description of the motivation for this analysis with this dataset. The comment on estimating the SNP-based Heritability of BP with other CV risk factors comes a little left field from the prior discussion of GWAS results. I was not able to follow in the written text the work you had done relating to the different ethnic groups.
5. The power of GWAS for different sample sizes across ancestries again requires some motivation text for this analysis – these results are not so well integrated following on from describing the results from a GWAS and inter-ethnic heterogeneity.
6. To note the FTO association with BP has recently been reported in a genome-wide association meta-analysis incorporating gene-smoking interactions for BP associated loci, this result should be commented upon and the reference included.

Reviewer #2 (Remarks to the Author):

Takeushi et al. describe a blood pressure discovery GWAS using large Asian and European studies (130,777 Asians in discovery and 289,038 Europeans in replication) and the authors claim 19 novel BP loci. Additionally the authors find: genetic heterogeneity between East Asians and Europeans, describe causal variant effect size correlations between Asians and Europeans, and provide an explanation for some of the ancestry-specific variants observed.

The subject is interesting, the text is well written, the analysis approaches are valid. The conclusions partly valid.

Strengths:

- 1) Detailed genetic analysis using a very large collection of Asian samples.
- 2) Data on the differences of the genetic origins of BP between Asians and Europeans.

Weaknesses:

- 1) This reviewer does not believe that the proof for the novelty of the 19 BP loci is sufficient. The cutoff for follow-up was very lenient ($<1 \times 10^{-5}$) and a large number of SNPs was taken forward. I would like to see how many SNPs replicate when correcting for the number of tests in replication. The text should inform the reader on the main steps of the analysis, without having to read the supplement. Main QC results such as GC and GC correction steps should be in the main text. The QQ plots should be presented without and with subtraction of the findings from the main SNPs.
- 2) The work on inter-ethnic heterogeneity, heritability, and ancestry-specific work has to make more clear (in the main text) which SNP-set was used for these analyses. What I would like to know is how many of the current BP SNPs has significant heterogeneity, this seems partially addressed in the section on ancestry-specific variants, but not for the other sections and the number of previously identified BP SNPs is not transparently explained (e.g. with a supplementary table). The sections on the inter-ethnic work may be summarized with a single quantitative statement that should also be included in the abstract.

We are very grateful for the reviewers' helpful comments.

Responses to reviewer 1's comments

1. This paper initially describes a GWAS for BP phenotypes, expanding on this groups earlier work on BP discovery and reports 19 new BP loci. The authors then go onto describe inter-ethnic heterogeneity of the signals at some BP loci (there are some novel observations here, although one of the genome-wide significant loci was the topic of some of this groups prior work). Up until this point the paper is easy to follow and interpret the findings, although the section on ancestry-specific SNP loci should follow next.

>> We have placed the section on 'Ancestry-specific SNP loci' at the back of the section on 'Interethnic heterogeneity of GWAS results' (Page 10) as the reviewer recommended.

2. The analyses to delve into ancestry specific loci for BP and other traits is not so well described and easy to follow. The authors need to define their motivation for pursuing the additional analyses and across traits and make their observations more succinct, including the power analyses.

>> We have revised the part of 'Genetic correlation and power of GWAS' as follows. First, we have described our motivation for pursuing the additional analyses and across traits: (1) "In the present study, the availability of genome-wide association data from >100,000 individuals for both East Asians and Europeans separately motivated us to perform additional analyses of systematic, genome-wide interethnic comparison" (in the second paragraph, Page 11); (2) "To estimate the degree of interethnic overlap and non-overlap of blood pressure loci" and "Similar to Europeans, the recent progresses of GWAS in East Asians motivated us to investigate different sample sizes in preparation for much-larger trans-ancestry meta-analysis" (in the third paragraph, Page 11); and (3) "We extended the interethnic analyses to other complex traits such as plasma lipid level, anthropometric measurement and type 2 diabetes using published GWAS summary statistics of relatively large number of samples (**Supplementary Table 14**)" (in the fourth paragraph, Page 11). Also, we have summarized our observations in the power analysis in the additional sentence, "As the sample sizes in both ethnic groups become larger, we can expect a higher proportion of interethnic overlap; nevertheless, more than or nearly half of the genome-wide significant loci may not overlap between the ethnic groups for GWAS of the same sample size" (in the first paragraph, Page 12).

3. At the moment the paper is a bit of a mix of topics, it may work better to describe all the work on BP then compare to results from other traits and then make the statements on common-ancestry specific association model.

>> We have revised the corresponding results section such that we describe all the work on BP in the first place (from Page 7 to the third paragraph in Page 11), followed by comparison with results from other traits (from the fourth paragraph in Page 11) and then proceed with discussion of “a common ancestry-specific variant association model” plus selective sweep at ancestry-specific loci (Pages 12 & 13) in the Results section, as suggested by reviewer 1.

Minor comments

1. Can the authors indicate which imputation panel was used for the discovery analyses, this is not stated in the text. I note the information is in Suppl. Table 2.

>> We have indicated the corresponding information in the section on ‘Populations and genotyping’ of the Online Methods.

2. Sentinel variants in a combined meta-analysis of discovery and replication samples reaching $P < 5 \times 10^{-8}$ are reported as novel loci in this study – can this definition be stated in the first paragraph of the results so the study design is totally clear in the text alongside the supplementary figure.

>> We have added the corresponding information to the first paragraph of the Results (Page 8).

3. ST6 presents the results from eQTL analyses, there is no text provided on the exact method and any indication of the coincidence of the BP SNP and top eQTL for the gene indicated? From a brief look at the results I do not think many of the BP SNPs are in high LD ($r^2 > 0.8$) with the top eSNP for some of the genes listed. The eQTL results should be carefully reviewed for presentation.

>> We have added the corresponding information to the section on ‘Functional annotations and candidate gene identification’ of the Online Methods (Page 25) and have indicated the coincidence (or LD coefficient) between the BP sentinel SNP and top eQTL for the gene in Supplementary Table 6 (which is renumbered as Supplementary Table 7 in the revised manuscript). Also, we have revised the description of eQTL results in the Results accordingly

(Page 8).

4. The section entitled genetic correlation and power of GWAS requires a better description of the motivation for this analysis with this dataset. The comment on estimating the SNP-based Heritability of BP with other CV risk factors comes a little left field from the prior discussion of GWAS results. I was not able to follow in the written text the work you had done relating to the different ethnic groups.

>> We have revised the part of 'Genetic correlation and power of GWAS' accordingly. Please see our responses to reviewer 1's comment #2.

5. The power of GWAS for different sample sizes across ancestries again requires some motivation text for this analysis – these results are not so well integrated following on from describing the results from a GWAS and inter-ethnic heterogeneity.

>> We have revised the part of 'Genetic correlation and power of GWAS' accordingly; that is, "Similar to Europeans, the recent progresses of GWAS in East Asians motivated us to investigate different sample sizes in preparation for much-larger transethnic meta-analysis" (in the third paragraph, Page 11). Please see our responses to reviewer 1's comment #2.

6. To note the FTO association with BP has recently been reported in a genome-wide association meta-analysis incorporating gene-smoking interactions for BP associated loci, this result should be commented upon and the reference included.

>> We have revised the corresponding part in the Results (the first paragraph in Page 9) and included the reference (#13, Sung YJ *et al.*).

Responses to reviewer 2's comments

Weaknesses:

1-1. This reviewer does not believe that the proof for the novelty of the 19 BP loci is sufficient. The cutoff for follow-up was very lenient ($<1 \times 10^{-5}$) and a large number of SNPs was taken forward. I would like to see how many SNPs replicate when correcting for the number of tests in replication.

>> SNPs attaining $P < 1 \times 10^{-5}$ in stage 1 consisted of 281 loci, of which 172 loci were previously unreported. Bonferroni's correction for 172 loci may yield a significance level of $0.05/172 = 0.00029$. Here, two loci (rs3853476 and rs10821808) identified in our meta-analysis combining

two ethnic groups may be excluded because they showed significant ($P < 0.01$) BP association in the additional European replication stage alone. Among the 17 (19 minus 2) loci, 5 loci satisfy this threshold in East Asian replication stage (stage 2), i.e., one-sided $P < 0.00029$, imposing concordant direction. However, we consider the other 12 loci to show statistically significant BP association as well, for the following reason.

For two-staged GWAS, there are generally two types of study design, namely, replication-based analysis and joint analysis [Nat Genet (2006) 38:209]. In the replication-based analysis, as described by the reviewer, evidence of replication mainly concentrated on stage 2 results, using a significance level of $0.05/(\text{the number of markers tested in stage 2})$. On the other hand, in the joint analysis, genome-wide significance is assessed by applying $P < 5 \times 10^{-8}$ to the combined results for both stages after meta-analysis.

In the above-cited article, joint analysis is recommended as it almost always results in increased power compared to replication-based analysis for the same type I error rate. We have thus adopted the joint analysis design in the current study. Several joint BP meta-analyses have examined whether the SNPs identified via GWAS can retain genome-wide significance (e.g., $P < 5 \times 10^{-8}$) after combining the additional stage samples with the discovery stage samples (e.g., Nat Genet 2016, ng.3667; Nature 2011, nature10405). There are BP GWAS meta-analyses, which have adopted an approach combining two types of study design. For example, $P < 5 \times 10^{-8}$ in combined meta-analysis plus $P < 0.05$ or $P < 0.01$ in replication data with the same direction of effect were adopted (e.g., Nat Genet 2016, ng.3654; Nat Genet 2017, ng.3768). When we apply this combined criteria ($P < 5 \times 10^{-8}$ in combined meta-analysis plus $P < 0.05$ with the same direction of effect) to 12 loci in our results, all but one (rs66658258) loci have satisfied them. For rs66658258, although the strength of association in East Asian replication stage is borderline significant ($P=0.056$) for MAP, it is nominal significant ($P=0.034$) for DBP, and $P < 5 \times 10^{-8}$ is attained in the combined samples for both BP traits ($P=4.6 \times 10^{-9}$ and 5.0×10^{-9} for MAP and DBP, respectively) (as shown in ST3 and ST5), satisfying the combined criteria.

Nevertheless, understanding the reviewer's concern, we have further looked up European GWAS results for the novel loci as demonstrated in ST4 (a new table). The European GWAS results were available at 6 of 17 loci that were identified via East Asian meta-analysis, as not all the corresponding sentinel SNPs are included in HapMap SNPs, with which European ICBP data were imputed. In all cases, the direction of effect is consistent between the ethnic groups and nominal significant association ($P < 0.05$) is detectable in Europeans for half (3) of the loci:

P=0.0086 at rs17622152, P=0.012 at rs9303509 and $P=3.9 \times 10^{-4}$ at rs6021247.

Consequently, we consider that the joint analysis controls type I error and is well powered to detect true associations in the present study.

1-2. The text should inform the reader on the main steps of the analysis, without having to read the supplement. Main QC results such as GC and GC correction steps should be in the main text. The QQ plots should be presented without and with subtraction of the findings from the main SNPs.

>> We have revised the corresponding part in the Results (last 4 lines, Page 7) and Supplementary Fig. 2.

2-1. The work on inter-ethnic heterogeneity, heritability, and ancestry-specific work has to make more clear (in the main text) which SNP-set was used for these analyses. What I would like to know is how many of the current BP SNPs has significant heterogeneity, this seems partially addressed in the section on ancestry-specific variants, but not for the other sections and the number of previously identified BP SNPs is not transparently explained (e.g. with a supplementary table).

>> We have made clear that “we used transethnic SNP data available for both East Asian ($N = 130,777$ from stage 1) and European (max $N = 105,253$ from ICBP and iGEN-BP) GWAS results in the subsequent analyses of interethnic heterogeneity” at the beginning of the section named ‘Interethnic heterogeneity of GWAS results’ (Page 9). Also, we have added a section named ‘Interethnic heterogeneity at common variant loci’ in both the Results (Pages 10 and 11) and Online Methods (Page 27) plus new supplementary materials (Supplementary Table 13 and Supplementary Fig. 7), in which all the corresponding information and results are demonstrated.

2-2. The sections on the inter-ethnic work may be summarized with a single quantitative statement that should also be included in the abstract.

>> We have added to the abstract a sentence, “At 6 unique loci, distinct non-rare (or common) ancestry-specific variants co-localized within the same linkage disequilibrium block despite the significantly discordant direction of effects for the proxy shared variants between the ethnic groups”.

Reviewer #1 (Remarks to the Author):

The authors have addressed all of my concerns satisfactorily, thank you.

I don't think you need to state "motivated us" again in the paragraph entitled "Genetic correlation and power of GWAS" this can be re-phrased a little.

Reviewer #2 (Remarks to the Author):

This is a revision of the previously reviewed paper by Takeushi et al. The manuscript has two parts: de-novo discovery and trans-ethnic work.

Part one does still not satisfy this reviewer: the cleaning steps are unsatisfactory (GC not applied, not indicated in the main text), the definition of replication is not satisfactory.

Part two mainly concentrates on the newly discovered loci from part one for the first analyses, but this is, as I understand, only a small part of the overall number of known BP loci. The method for identifying causal variants is not satisfactory, although this is a central element for inter-ethnic comparisons.

This reviewer holds the view that the title chosen does not adequately describe the work and may even be misleading "Interethnic comparability in blood pressure loci".

We are very grateful for the reviewers' helpful comments. In the word files ('Main text' and 'Supplementary online material'), we highlighted the revised parts of the manuscript in yellow.

Responses to reviewer 1's comments

I don't think you need to state "motivated us" again in the paragraph entitled "Genetic correlation and power of GWAS" this can be re-phrased a little.

>> We have re-phrased the corresponding part of the manuscript; "the recent progresses of GWAS in East Asians prompted us to investigate ~ (line 11-12 in Page 14)."

Responses to reviewer 2's comments

Part one does still not satisfy this reviewer:

1. the cleaning steps are unsatisfactory (GC not applied, not indicated in the main text)

>> "Genomic control and intercepts from linkage disequilibrium (LD) Score regression were calculated at each study level ($\lambda_{GC} = 0.89-1.24$ and LD Score regression intercept = $0.94-1.06$), indicating no apparent confounding biases such as population stratification (**Supplementary Table 2**). Since the LD Score regression intercept can account for polygenic effects and inflation due to large sample size (Bulik-Sullivan *et al.* Nat Genet 2015), we applied the LD Score regression intercept as a correction factor for cohorts with a sample size of >3,000 individuals. Genomic control λ_{GC} was used as a correction factor in the other studies." Here, LD Score regression is unsuitable for cohorts with a sample size <3,000 individuals

(https://www.med.unc.edu/pgc/statgen/presentations/pgc_stat_bulik_2015.pdf).

Table. Genotyping and correction factor in the stage 1 cohorts (See details in ST 2)

Study name	Sample size for BP	Genotyping platform	Genomic control lambda (LD score regression intercept)				
			DBP	SBP	MAP	PP	HT
Stage 1 cohorts							
BBJ	125,778	Illumina	1.18	1.24	1.22	1.15	1.22
		HumanOmniExpressExome OR IlluminaHumanOmniExpress + IlluminaHumanExome	(1.05)	(1.06)	(1.06)	(1.04)	(1.04)
CAGE- Amagasaki	559	Illumina Omni2.5	1.00 (1.00)	0.99 (1.01)	0.99 (1.00)	0.97 (1.01)	1.12 (1.00)

CAGE-GWAS1	1547	Illumina 550K/610K	1.01 (0.99)	1.02 (1.00)	1.01 (0.99)	1.00 (1.01)	1.04 (1.00)
CAGE-KING- Omni2	527	Illumina Omni2.5-8	1.01 (0.99)	1.01 (0.99)	1.01 (0.99)	1.01 (0.99)	0.98 (1.00)
CAGE-KING- OmniE1	817	Illumina OmniExpress-12	0.99 (1.02)	0.99 (1.01)	0.99 (1.02)	0.98 (1.00)	0.97 (1.02)
CAGE-KING- OmniE2	615	Illumina OmniExpress-24	1.00 (1.01)	1.00 (1.00)	1.00 (1.00)	1.00 (0.99)	0.98 (1.00)
CAGE-KING- Quad	494	Illumina 660W-Quad	1.00 (0.99)	1.00 (0.98)	1.00 (0.98)	1.00 (0.99)	NA
AASC	448	Illumina Omni2.5-4	1.02 (1.01)	1.01 (1.00)	1.02 (1.01)	1.01 (1.00)	0.89 (0.94)

We have described these in both the main text (last 4 lines in Page 7 and first 4 lines in Page 8) and the online methods (the first paragraph in Page 27). We have also added the results for λ_{GC} and LD Score regression intercept to **Supplementary Table 2**. We have repeated the cleaning steps in the discovery samples and found that at two loci (rs4418728 on chromosome 10 and rs1078967 on chromosome 15), the strength of association was attenuated in the meta-analysis of East Asian stage 1+2 but reached a genome-wide significance level in transethnic meta-analysis (ST3 and ST4). We have revised all the related data in Table 1, Supplementary Tables (**ST3-13** and **ST15**) and Figures (**SF1-4**) accordingly.

2. the definition of replication is not satisfactory

>> We performed additional lookups in European-descent samples, including large-scale data sets for the UK Biobank blood pressure (SBP and DBP) GWAS ($N =$ up to 422,771), which are publicly available via <https://doi.org/10.1038/s41588-018-0144-6> and <http://www.nealelab.is>. We tested blood pressure associations in three independent data sets: discovery stage, replication stage and lookups, denoting the meta-analysis of discovery and replication stages as the "combined meta-analysis".

We have revised the definition of a validated association signal as follows.

"In the present study, an association signal was declared to be validated if it satisfied all four of the following criteria: (i) the sentinel SNP was genome-wide significant ($P < 5 \times 10^{-8}$) in the combined meta-analysis for any of the five blood pressure traits; (ii) the sentinel SNP showed evidence of support ($P < 0.05$) in the replication stage alone for association with the most significantly associated blood pressure trait from the

combined meta-analysis; (iii) the sentinel SNP showed further evidence of support ($P < 0.05$) in association results for either SBP or DBP of lookup variants ($n = 18$ in this study); and (iv) the sentinel SNP had concordant directions of effect across the discovery and replication stages and the lookups. ”

Table. Revised summary data for BP association of 19 sentinel SNPs (See details in ST 4)

Sentinel SNP in the locus		Discovery (EAS)			Replication		Combined (discovery + replication)		Lookup (EUR)		
Position	SNP	EAF	P	N	P	N	P	N	EAF	P	N
1: 27972058	rs2076460	0.30	3.2E-06	130777	2.2E-04	44069	3.6E-09	174846	0.00		
1: 155190254	rs2990220	0.83	1.5E-08	130777	3.3E-05	52877	2.2E-12	183654	0.51	4.3E-02	317754
3: 46896499	rs6772151	0.29	1.8E-06	127876	9.4E-04	28627	7.8E-09	156503	0.07	2.9E-02	80124
3: 183520112	rs17622152	0.47	2.7E-06	130777	2.1E-03	52982	2.0E-08	183759	0.33	9.9E-07	422771
6: 1621042	rs12209106	0.68	1.4E-06	130777	1.0E-03	29659	6.4E-09	160436	0.60	8.9E-05	422771
7: 1141470	rs78399431	0.24	8.8E-07	130777	3.1E-03	48634	9.6E-09	179411	0.25	2.4E-06	422771
10: 48434420	rs2125067	0.12	6.1E-07	130777	2.0E-03	48226	4.8E-09	179003	0.02	2.9E-04	422771
11: 120340060	rs2305013	0.85	4.0E-08	130777	3.6E-03	50117	5.6E-10	180894	0.95	2.1E-05	422771
12: 32692233	rs5006548	0.16	4.9E-07	50786	1.2E-02	21061	2.2E-08	71847	0.35	1.3E-03	422771
14: 100793431	rs1535464	0.10	5.0E-05	130777	8.0E-06	52913	3.5E-09	183690	0.25	1.6E-03	422771
16: 53802494	rs11642015	0.21	3.9E-07	130777	4.6E-08	44140	1.9E-12	174917	0.40	7.2E-05	317754
17: 64530887	rs9303509	0.40	3.1E-06	130777	3.0E-04	52992	3.9E-09	183769	0.34	1.2E-02	104915
19: 18455657	rs66978877	0.55	2.9E-08	50786	3.9E-02	18064	4.5E-09	68850	0.73	1.0E-11	422771
20: 50108980	rs6021247	0.58	1.8E-06	130777	7.0E-04	53008	5.0E-09	183785	0.53	2.6E-12	422771
20: 61462502	rs66658258	0.58	6.9E-08	130777	3.4E-02	33861	1.0E-08	164638	0.07	2.9E-01	422771
5: 141817754	rs3853476	0.58	1.3E-05	130777	7.7E-05	114054	6.0E-09	244831	0.37	2.0E-04	422771
10: 62390646	rs10821808	0.58	2.8E-05	130777	2.8E-05	158140	3.4E-09	288917	0.50	1.3E-04	422771
10: 94839724	rs4418728	0.62	8.7E-06	130777	3.9E-04	125341	1.5E-08	256118	0.45	1.3E-04	422771
15: 74222987	rs1078967	0.15	1.3E-07	130777	2.6E-03	134504	5.6E-09	265280	0.15	1.0E-06	422771

With the exception of rs2076460, which is very rare in Europeans (MAF = 0.002 in 1000 Genomes EUR), and rs66658258, 17 sentinel SNPs showed nominal significant ($P < 0.05$) blood pressure association with the concordant direction of allelic effects

(**Supplementary Table 4**) in lookups, thus validating the loci.

We have added these to the online methods (last 2 lines in Page 27 and the first and second paragraphs in Page 28) and revised the main text (line 21-26 in Page 8). The results for the replication stage and lookups are shown in Supplementary Table 4. Based on the revised definition, we report that 17 of the 19 newly identified SNP loci have been validated in this study. The main text, Table 1 and other parts of the manuscript have been revised accordingly.

3. Part two mainly concentrates on the newly discovered loci from part one for the first analyses, but this is, as I understand, only a small part of the overall number of known BP loci. The method for identifying causal variants is not satisfactory, although this is a central element for inter-ethnic comparisons.

>> As pointed out by the reviewer #2, we have revised the part of interethnic comparisons to a large extent. First of all, we have clearly described the tested BP loci in reference to total non-rare blood pressure loci that have been previously reported or newly identified in the present study, by addition of the corresponding paragraph in the main text (in the section entitled “Ancestry-specific SNP loci”, Page 10), **Supplementary Fig. 7** and the section entitled “Interethnic heterogeneity at non-rare variant loci” in the online methods (Page 30-31).

A total of 750 previously reported SNPs (listed in **Supplementary Table 6**) plus 19 newly identified SNPs could be classified into 485 loci by regarding two SNPs at most 500 kb apart to belong to the same locus. After exclusion of 39 loci (MAF < 0.01 in both East Asians and Europeans, or no data available in GWAS data sets for both populations), 446 loci were retained and categorized into two groups—group 1 and group 2. Group 1 consisted of 382 loci with MAF \geq 0.01 in both populations and group 2 consisted of 64 loci with potential ethnic specificity, i.e., MAF < 0.01 in either East Asians or Europeans. Group 2 was further classified into group 2a (46 loci with MAF < 0.01 in one population and MAF \geq 0.05 in the other) and group 2b (18 loci with MAF < 0.01 in one population and $0.01 \leq$ MAF < 0.05 in the other). Since ICBP and iGEN-BP (European replication sample) data were imputed with HapMap SNPs, approximately one-third of group-1 SNPs were unavailable in our European GWAS data sets. Thus, 242 (out of 382) loci in group 1 were subjected to interethnic comparison of genetic impact on a lead blood pressure trait (**Supplementary Fig. 7a** and **Supplementary Table 13**).

Supplementary Fig. 7. Investigation of interethnic heterogeneity at blood pressure loci previously reported and newly identified.

By calibrating the proportion in this group-1 subset, we estimated the proportion of loci showing significant interethnic heterogeneity within the overall blood pressure loci tested ($N = 446$). The estimated proportion was 2.5% each in group 1 and group 2a (**Supplementary Fig. 7b**) as follow.

In the present study, interethnic heterogeneity of genetic impact on blood pressure was tested largely via two approaches, that is, for group-2a SNPs (46 + 2 SNPs) and group-1 SNPs (242 SNPs), respectively. For the first approach, we have described more detailed findings and explanations in the main text (the section entitled “Ancestry-specific SNP loci” in Page 10-12), **Supplementary Figs. 8 and 9**, and the online methods (the section entitled “Exploration of transeethnic haplotype SNPs from ancestry-specific SNPs” in Page 29-30). For the second approach, we also have described more detailed findings and explanations in the main text (the section entitled “Interethnic heterogeneity at variants polymorphic in both ancestries” in Page 12-13), **Supplementary Figs. 7, 10 and 11**, and the online methods (the section entitled “Interethnic heterogeneity at non-rare variant loci” in Page 30-31, above-mentioned).

Furthermore, we have revised the relevant parts of Discussion (in Page 17-18) accordingly.

For group-2b SNPs with potential ancestry specificity (18 SNPs in **Supplementary Fig. 7a**; $MAF < 0.01$ in one population and $0.01 \leq MAF < 0.05$ in the other), we did not investigate interethnic heterogeneity of association signals because of difficulties in the relevant test for rare ($MAF < 0.01$) and low-frequency ($0.01 \leq MAF < 0.05$) genetic variants by using imputed GWAS results (see ref. 17). We have mentioned this in the main text (the second paragraph in Page 12). Despite such limitations inherent to imputed GWAS results, we could successfully identify, in this study, 10 unique blood pressure loci showing significant interethnic heterogeneity that is attributable to ancestry-specific variants (e.g., at *C10orf107* and *CACNB2*; **Supplementary Fig. 9** and **Supplementary Table 13**), thereby providing evidence for the reduced genome-wide correlation of causal-variant effect sizes at SNPs common in two ancestry groups; i.e., we show that the genetic correlations in SBP and DBP between two ancestries are 0.898 (SE 0.040) and 0.851 (SE 0.046) respectively, and significantly different from 1 ($P = 0.005$ for SBP and $P = 0.0007$ for DBP).

4. This reviewer holds the view that the title chosen does not adequately describe the work and may even be misleading "Interethnic comparability in blood pressure loci".
>> We have revised the title as "Interethnic analyses of blood pressure loci in populations of East Asian and European".

Reviewer #2 (Remarks to the Author):

The authors have provided an extensive and detailed revision that is much appreciated.

In the view of this reviewer the evidence for replication in BP-SNP discovery is still very weak - the authors should go beyond nominal p-values and account for the number of SNPs tested.

We are very grateful for reviewer 2's helpful comments. In the word files ('Main text' and 'Supplementary online material'), we highlighted the revised parts of the manuscript in yellow.

Responses to reviewer 2's comments

In the view of this reviewer the evidence for replication in BP-SNP discovery is still very weak - the authors should go beyond nominal p-values and account for the number of SNPs tested.

>> To enhance the power and stringency of replication, we newly included lookup results from the China Kadoorie Biobank ($N = 94,201$) and also applied Bonferroni's correction for multiple testing to our replication study. By adopting a joint analysis strategy (Skol *et al.* Nat Genet 2006: ref. 12), we have made small revisions to the study design as below (**Supplementary Fig. 1**) with a GWAS (which consists of stages 1 and 2, $N =$ up to 289,038) followed by a replication study ($N = 516,972$).

All genome-wide SNP markers (6.2 million SNPs) are characterized in a proportion of the GWAS samples, that is, stage 1 ($N = 130,777$), and results of stage 1 are used to select a proportion of the SNP markers for follow-up (a list of 13,003 SNPs in this study) on the remaining GWAS samples, that is, stage 2 ($N = 53,008$ for EAS alone and $N = 158,261$ for EAS+EUR). In this joint analysis strategy, test statistics from stages 1 and 2 are combined by meta-analysis, and a genome-wide significance level ($P < 5 \times 10^{-8} =$

0.05/10⁶) can be used to claim significant association after adjustment for multiple testing of 10⁶ markers that are in linkage equilibrium in theory. In the current study design we further examined sentinel novel SNPs ($N = 19$) that reached $P < 5 \times 10^{-8}$ in the GWAS meta-analysis in an independent replication panel using the Bonferroni-corrected significance level ($P < 0.00263 = 0.05/19$), as suggested by reviewer 2. Moreover, to confirm robustness of BP-SNP association at the loci tested in replication by increasing statistical power, we used a new East Asian data set from the China Kadoorie Biobank ($N = 94,201$) in addition to a previous European data set from the UK Biobank ($N = 422,771$). Consequently, with the exception of 4 SNPs, 15 sentinel SNPs showed significant ($P < 0.00263$) blood pressure association in a replication study with the concordant direction of allelic effects (**Supplementary Table 4**), thus validating the loci.

We have revised the part of '*Genome-wide association analyses and lookups for replication*' in the Results (Pages 7 and 8), as well as Discussion (first 3 lines in Page 16) and the part of '*GWAS and replication meta-analyses*' in the online methods accordingly. Also, we have revised all the related data and materials in Table 1, Supplementary Table 4, Supplementary Table legends and Supplementary Fig. 1, and have added three references (#12, 14 and 15) to the text. We have put collaborators from the China Kadoorie Biobank to the co-author list.

Reviewer #2 (Remarks to the Author):

Reasonable revision. The effort of additional replication is much appreciated.